# Selenium-alloyed tellurium oxide for amorphous p-channel transistors

Ao Liu[1,2,3 ✉], Yong-Sung Kim[4,5], Min Gyu Kim[6], Youjin Reo[2], Taoyu Zou[2], Taesu Choi[2], Sai Bai[1], Huihui Zhu[2,3,7 ✉] & Yong-Young Noh[2 ✉]

Compared to polycrystalline semiconductors, amorphous semiconductors offer inherent cost-effective, simple and uniform manufacturing. Traditional amorphous hydrogenated Si falls short in electrical properties, necessitating the exploration of new materials. The creation of high-mobility amorphous n-type metal oxides, such as a-InGaZnO (ref. 1), and their integration into thin-film transistors (TFTs) have propelled advancements in modern large-area electronics and new-generation displays[2–8]. However, finding comparable p-type counterparts poses notable challenges, impeding the progress of complementary metal–oxide–semiconductor technology and integrated circuits[9–11]. Here we introduce a pioneering design strategy for amorphous p-type semiconductors, incorporating high-mobility tellurium within an amorphous tellurium suboxide matrix, and demonstrate its use in high-performance, stable p-channel TFTs and complementary circuits. Theoretical analysis unveils a delocalized valence band from tellurium 5p bands with shallow acceptor states, enabling excess hole doping and transport. Selenium alloying suppresses hole concentrations and facilitates the p-orbital connectivity, realizing high-performance p-channel TFTs with an average field-effect hole mobility of around 15 cm$^2$ V$^{-1}$ s$^{-1}$ and on/off current ratios of 10$^6$–10$^7$, along with wafer-scale uniformity and long-term stabilities under bias stress and ambient ageing. This study represents a crucial stride towards establishing commercially viable amorphous p-channel TFT technology and complementary electronics in a low-cost and industry-compatible manner.

Creating high-mobility amorphous p-type oxide semiconductors holds the promise of enhancing scalable complementary metal–oxide–semiconductor (CMOS) technology and facilitating the integration of multifunctional electronics. However, the current hurdle lies in the highly localized valence band maximum (VBM) states, consisting of anisotropic oxygen 2p orbitals. In conventional p-type oxides such as Cu$_2$O and SnO, the valence band orbital hybridization imparts decent p-type characters whereas the device performance remains constrained, even with the crystalline channel[11–15]. Whereas using amorphous hydrogenated Si for cost-effective, large-area production is a viable option, its low field-effect hole mobility ($\mu_h$ < 0.1 cm$^2$ V$^{-1}$ s$^{-1}$) restricts its modern applications. Benefiting from the high mobility and stability, low-temperature polycrystalline silicon is combined at present with n-type oxides for complementary circuit and display backplane applications. Nevertheless, this use is confined to small- and/or medium-area devices due to the intricate process flow, inhomogeneity from the grain boundary and challenges in upscaling mass production[16]. Extensive efforts have been directed towards exploring organic compounds[17,18], metal halides[19–22] and low-dimensional nanomaterials[23–29] as p-type semiconductors for transistors. However, these materials show optimal performance only in crystallized form and they come with intrinsic limitations such as low stability, complex synthesis processes, large-area non-uniformity and a lack of industrial compatibility.

In this study, we propose an alternative route to designing amorphous p-type semiconductor, which involves a mixed phase of high-mobility tellurium within an amorphous tellurium suboxide matrix (Te–TeO$_x$, 0 < x ≤ 2). The thermal evaporation method was used to deposit amorphous Te–TeO$_x$ thin films by evaporating tellurium dioxide (TeO$_2$) powder, followed by low-temperature annealing under ambient conditions at 225 °C. For the deposition of Se-alloyed Te–TeO$_x$, Se was blended with TeO$_2$ powder before the evaporation (more details in the Methods). The Se addition had no impact on the microstructure, nevertheless, it was instrumental in improving the electrical properties (which we discuss later). The X-ray diffraction (XRD) patterns show the typical amorphous features of deposited films before and after thermal annealing (Fig. 1a). High-resolution transmission electron microscopy (HRTEM) and diffraction analyses confirmed the amorphous-like nature, showing no perceptible crystalline domains or long-range orders (Fig. 1b,c). This amorphous and short-range disorder microstructure aligns with previous observations on evaporated tellurium oxide[30–32]. The Te K-edge X-ray absorption near edge structure (XANES) spectrum of the deposited film shows characteristics resembling a mixture

[1]Institute of Fundamental and Frontier Sciences, University of Electronic Science and Technology of China, Chengdu, China. [2]Department of Chemical Engineering, Pohang University of Science and Technology, Pohang, Republic of Korea. [3]Department of Chemistry, Northwestern University, Evanston, IL, USA. [4]Korea Research Institute of Standards and Science, Daejeon, Republic of Korea. [5]Department of Nano Science, University of Science and Technology, Daejeon, Republic of Korea. [6]Beamline Research Division, Pohang Accelerator Laboratory, Pohang University of Science and Technology, Pohang, Republic of Korea. [7]School of Physics, University of Electronic Science and Technology of China, Chengdu, China. ✉e-mail: ao.liu@uestc.edu.cn; hhzhu@uestc.edu.cn; yynoh@postech.ac.kr

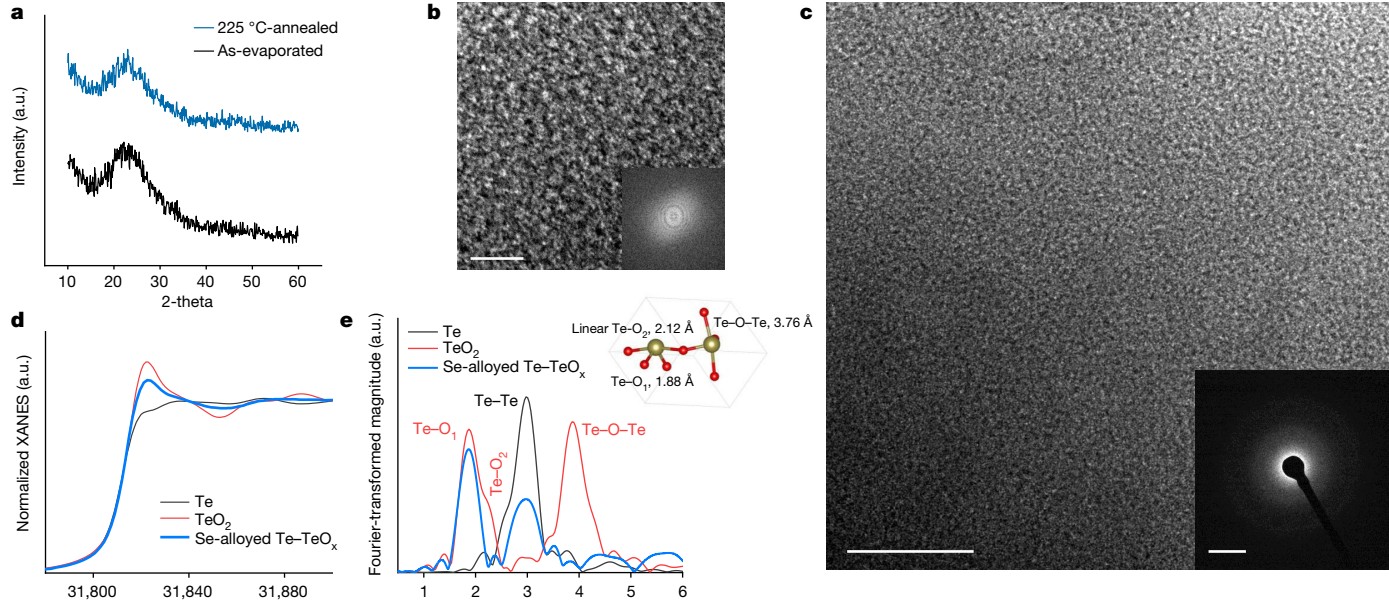

**Fig. 1 | Structural characterizations of amorphous Se-alloyed Te–TeO$_x$.**
**a**, XRD spectra of as-evaporated and 225 °C-annealed Se-alloyed Te–TeO$_x$ thin films on glass. **b**,**c**, HRTEM images, the fast Fourier transform spot patterns (**b**, inset) and selected area electron diffraction pattern of 225 °C-annealed Se-alloyed Te–TeO$_x$ (**c**, inset). **d**, Te K-edge XANES spectra of the Se-alloyed

Te–TeO$_x$ film and reference materials of elemental Te and TeO$_2$. **e**, Corresponding Fourier transform of Te K-edge $k^3$-weighted EXAFS spectra. The inset shows the tetragonal TeO$_2$ bonding model. Scale bars, 5 nm (**b**), 20 nm (**c**) and 51/nm (**c**, inset). a.u., arbitrary units.

of Te and TeO$_2$ reference spectral features (Fig. 1d). A linear combination analysis indicates an averaged Te/TeO$_2$ composition ratio of roughly 4/6 (detailed discussion in Supplementary Methods), and the atomic ratio of Te/O was estimated to be 1/1.2. Compared to reference TeO$_2$ with peak feature of shorter Te–O$_1$ and longer Te–O$_2$, the Fourier transform for extended X-ray absorption fine structure (EXAFS) of the deposited film clearly shows a slight decrease in the shorter Te–O$_1$ bond and the sharp diminishing of both longer Te–O$_2$ and Te–O–Te long-range ordering (Fig. 1e and Extended Data Fig. 1). The generation of oxygen vacancy breaks the bridged bond of Te–O$_2$, and the undercoordinated Te leads to the loss of Te–O–Te long-range ordering, forming the amorphous structure. Furthermore, noticeable metallic Te–Te bonds were observed, suggesting the spontaneous generation of elemental Te in the final films. The several components were further confirmed by X-ray photoelectron spectroscopy analysis (Extended Data Fig. 2). This can be related to the redox behaviour of tellurium, in which partial Te$^{4+}$ was reduced to elemental Te in molten TeO$_2$ in an inert atmosphere[32] and with the tungsten boat reaction[33]. Therefore, it is plausible that the composite film is composed of a mixed phase of Te–TeO$_x$.

As elemental Te is present, the likelihood of local Te nanocrystals cannot be ruled out. However, their embeddedness and dispersion within the amorphous TeO$_x$ matrix tend to induce short-range disordering. Furthermore, using an amorphous substrate at room temperature is beneficial to the formation of a highly disordered amorphous-like structure. The heavy dissociation of TeO$_2$ during evaporation could lead to the simultaneous impingement of Te and TeO$_x$ on the substrate and the limited mobility and mutual solubility of the adsorbed atoms on the substrate can result in condensation at or near the impingement point before reaching a more energetically favourable site.

Leveraging the insights gained from XANES and EXAFS results and the film density of 5.6 g per cm$^3$ obtained from X-ray reflectivity analysis, we conducted density functional theory (DFT) calculations to explore the energy band structure and electrical properties of amorphous Te–TeO$_x$. For stoichiometric amorphous TeO$_2$, the VBM primarily consists of localized O-2$p$ states, indicating the poor p-type character (Extended Data Fig. 3). This stands in marked contrast to recent studies

on orthorhombic low-dimensional β-TeO$_2$, which shows favourable p-type properties[34–36]. For non-stoichiometric amorphous Te–TeO$_x$, the radial distribution functions (RDFs) generated in DFT indicate the diminishment of long Te–O$_2$ bond (Fig. 2a). The generated atomic structure in Fig. 2b encompasses a variety of Te–Te bonds and undercoordinated (zero-, one-, two- and threefold) Te atoms with oxygen. The average coordination number of Te with oxygen is calculated to be 2.5. The electronic density of states illustrates that the VBM is dominated by partially occupied Te-5$p$ defect bands above the O-2$p$ states (Fig. 2c). The Te-5$p$ states, originating mainly from Te in Te–TeO$_x$, serve as the hole transport channel and act as shallow acceptors. The spatially dispersed and percolated Te-5$p$ orbitals throughout the amorphous network, along with enough Te in Te–TeO$_x$, contribute to the dispersed VBM. The charge densities of the Te-5$p$ defect band and the shallow acceptor state near Te-5$p$ defect band are illustrated in Fig. 2d,e, respectively. The theoretical bandgap of TeO$_{1.2}$ is 0.91 eV in HSE06 (0.49 eV in DFT-PBE (Perdew–Burke–Ernzerhof)), slightly lower than the experimental value of around 1.1 eV (Extended Data Fig. 4).

To assess the potential of Te–TeO$_x$-based semiconductors for electronic device applications, we fabricated bottom-gate, top-contact TFTs by depositing films on a 100 nm SiO$_2$ dielectric containing Ni source and drain electrodes. The transfer and output characteristics are presented in Fig. 3a,b. The pristine Te–TeO$_x$ TFT showed typical p-channel behaviour with an average $\mu_h$ of 4.2 cm$^2$ V$^{-1}$ s$^{-1}$ and an on/off current ratio ($I_{on}/I_{off}$) of around 10$^4$. The onset voltage showed a pronounced positive shift, indicating a relatively high hole concentration in the channel. The TFT performance notably improved with Se doping, evident from a reduced onset voltage, lowered off-state current and increased $\mu_h$. The optimized Se alloying atomic percentage was determined to be around 25% (Se/Te, 1/3) using high-resolution inductively coupled plasma mass spectrometry (Extended Data Fig. 5). The deposited Se-alloyed Te–TeO$_x$ TFTs delivered an average $\mu_h$ of around 15 cm$^2$ V$^{-1}$ s$^{-1}$ (forward scan; $\mu_h$ from reverse scan is around 14 cm$^2$ V$^{-1}$ s$^{-1}$), an $I_{on}/I_{off}$ of roughly 10$^7$ and onset voltages of 20–25 V, whereas higher Se alloying percentages were found to degrade the TFT performance with a notable n-doping effect. Similar to many semiconductors, the

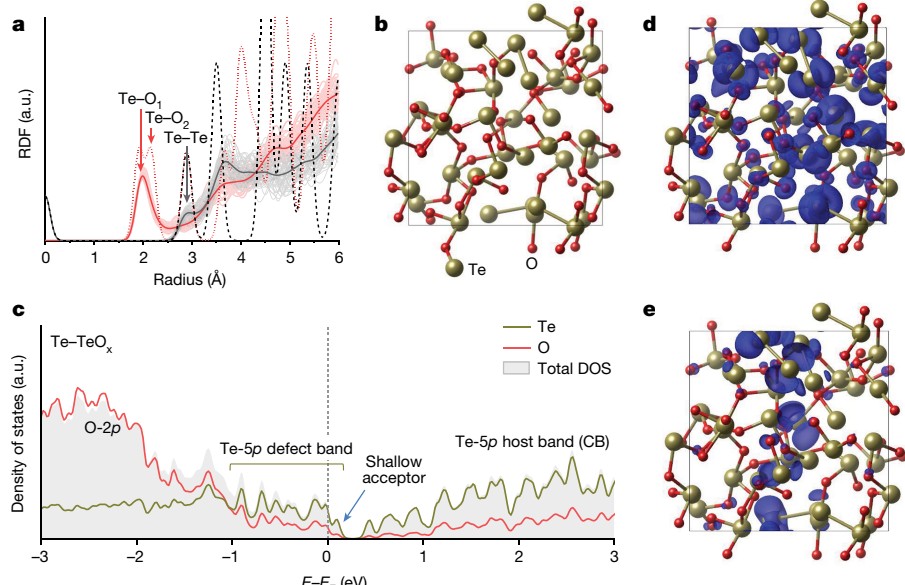

**Fig. 2 | Atomic and electronic structures. a**, Averaged Te–O (red) and Te–Te (black) RDFs of amorphous Te–TeO$_x$ (Te/O atomic ratio 1/1.2) generated in DFT. Light red and grey lines are the Te–O and Te–Te RDFs for all the 50 generated Te–TeO$_x$ samples; those for the crystalline TeO$_2$ and Te are shown with dashed lines. **b**, Atomic structure of amorphous Te–TeO$_x$ generated in DFT. **c**, Projected density of states (DOS) of the O-2$p$ (red) and Te-5$p$ (dark yellow) states in DFT-PBE. The Te-5$p$ host band comes from fourfold coordinated normal Te$^{4+}$ in TeO$_2$; the Te-5$p$ defect band mainly comes from elemental Te in Te–TeO$_x$. CB, conduction band. **d,e**, Charge density of the Te-5$p$ defect band (**d**) and the shallow acceptor state near the Te-5$p$ defect band (**e**) in amorphous Te–TeO$_x$.

channel layer thickness and postannealing temperature affected the TFT performance and, herein, the optimized Se-alloyed Te–TeO$_x$ channel thickness and annealing temperature are around 15 nm and 225 °C, respectively (Extended Data Fig. 6). We noted that an increased

applied $V_{DS}$ resulted in increased $I_{off}$ and a positively shifted onset voltage, especially at low Se alloying percentages. This behaviour may arise from the presence of narrow bandgap elemental Te. From the output curves, good current linearity (saturation) was observed at low (high)

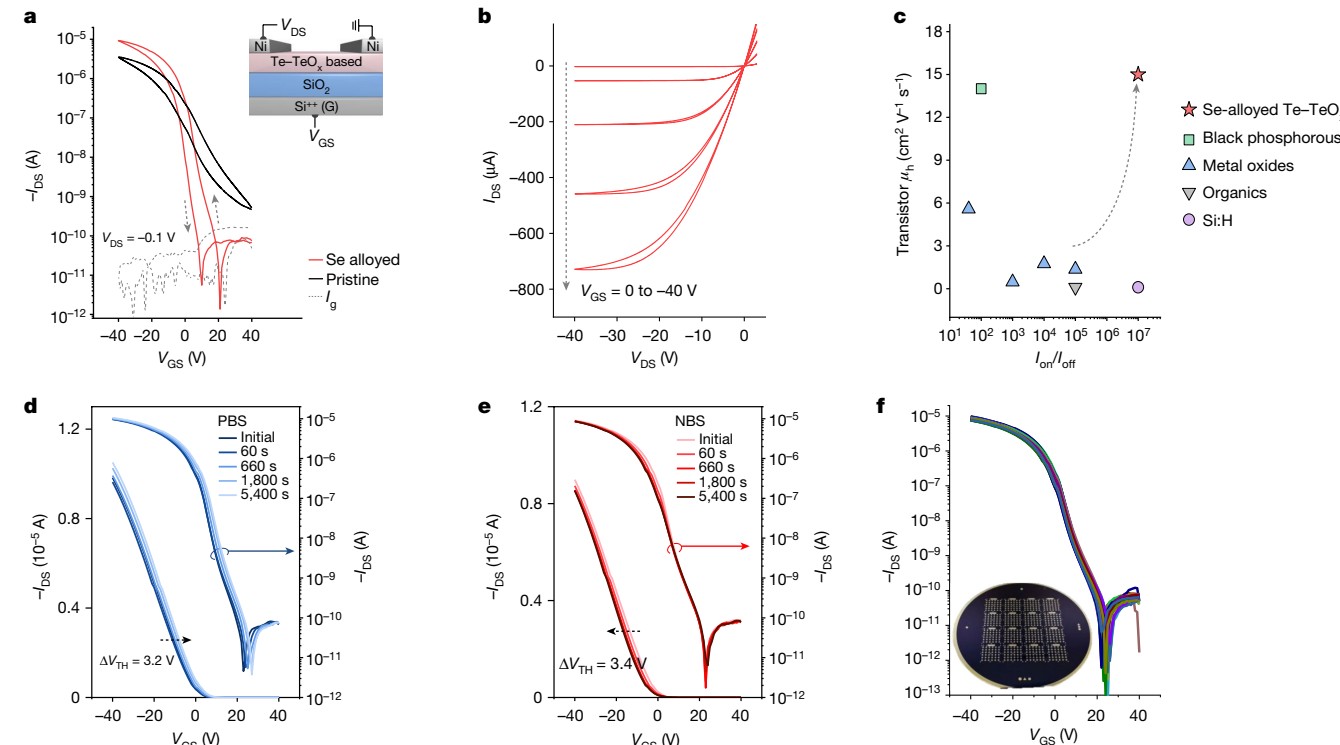

**Fig. 3 | Electrical characterizations of amorphous p-channel Se-alloyed Te–TeO$_x$ TFTs on a 100 nm SiO$_2$ dielectric. a**, Transfer characteristics of pristine Te–TeO$_x$ and Se-alloyed Te–TeO$_x$ TFTs; the inset shows TFT geometry (both hysteresis directions are counterclockwise). **b**, Output curves of one Se-alloyed Te–TeO$_x$ TFT. **c**, Benchmark of $\mu_h$ and $I_{on}/I_{off}$ of reported amorphous p-channel TFTs. **d,e**, Transfer curves and the $V_{TH}$ shifts of Se-alloyed Te–TeO$_x$ TFTs under positive bias stress (PBS) (**d**) and negative bias stress (NBS) (**e**) tests (±20 V) with different time durations. **f**, Transfer curves of 80 randomly measured TFTs fabricated by means of the optimized condition. The inset shows the optical image of TFT arrays on a 10 cm (4 inch) SiO$_2$ wafer.

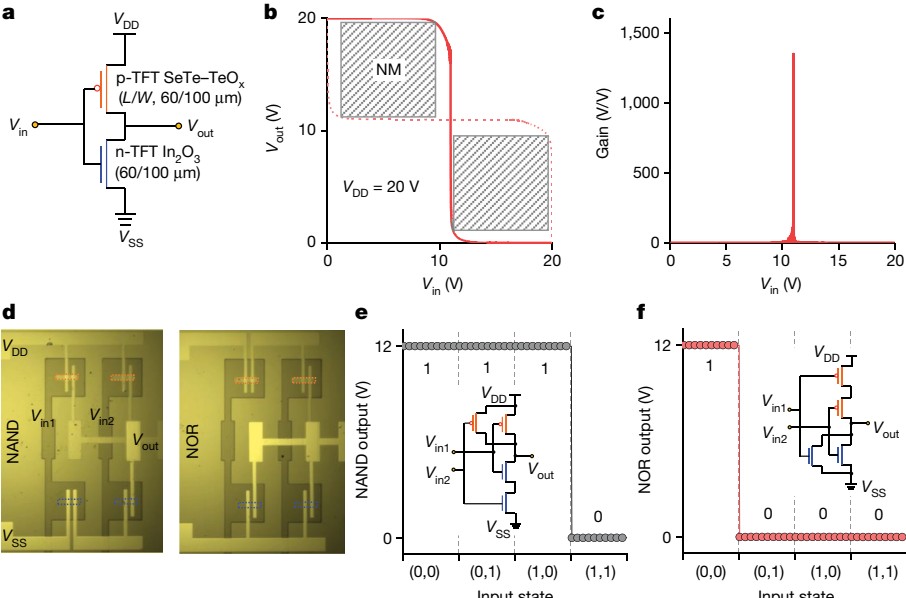

**Fig. 4 | Integrated CMOS circuits on a 100 nm HfO₂ dielectric. a–c**, Diagram (**a**), voltage transfer, noise margin (NM) extraction (**b**) and gain voltage curves (**c**) of one complementary inverter based on n-channel $In_2O_3$ and p-channel Se-alloyed $Te–TeO_x$ TFTs at a $V_{DD}$ of 20 V. **d–f**, Photograph (**d**), input and output waveforms for complementary NAND (**e**) and NOR (**f**) logic gates at a $V_{DD}$ of 12 V. The red and blue boxes in **d** indicate the position of p-channel Se-alloyed $Te–TeO_x$ and n-channel $In_2O_3$ TFTs, respectively. *L*, length; *W*, width.

source-drain voltages, indicating Ohmic contact between the channel layer and electrodes (Fig. 3b). A reasonably low contact resistance of 200 Ω cm was calculated using the transmission-line method[37]. The overall electrical performance surpasses that of p-channel TFTs based on various amorphous semiconductors, such as a-Si:H, organics and metal oxides (Fig. 3c and Extended Data Table 1).

Continuing our exploration, we delved into the Se state and its doping effect on the electrical properties of this amorphous hybrid system. EXAFS analysis confirmed the existence of Se in Se–Te metallic bonds, indicating that the Se was alloyed into the Te in $Te–TeO_x$ (Extended Data Fig. 7). Because of the deeper Se-4$p$ state compared to Te-5$p$, the Se alloy can reduce the number of empty 5$p$ acceptor states and thus the overall hole conductivity. The fully occupied Se-4$p$ orbitals in the valence band can facilitate the filled Te-5$p$-orbital connectivity as well throughout the amorphous network, rationalizing to the enhancement in hole mobilities. The corresponding DFT atomic and band structure are illustrated in Extended Data Fig. 8. Moving on to investigate the device operational stability performance, another critical metric for practical applications, we conducted constant bias stress tests on the Se-alloyed $Te–TeO_x$ TFTs. The results demonstrate decent operational stability, with threshold voltage ($V_{TH}$) shifts of 3.2 and 3.4 V observed after 5,400 s of positive and negative bias stress tests, respectively (Fig. 3d,e). The negligible variation in the subthreshold region indicated that the Se-alloyed $Te–TeO_x$ channel remained electrically robust during operation, with minimal generation of new defects; the primary instability was attributed to charge trapping[20]. Furthermore, the as-fabricated Se-alloyed $Te–TeO_x$ TFTs showed good ambient durability. Unlike conventional p-type oxide semiconductors, the main composition of metastable cations such as $Cu^+$ and $Sn^{2+}$ renders them sensitive to oxidation. Other emerging amorphous p-type semiconductors, such as halides and nanomaterials, often face susceptibility to air, impeding their practical applications.

Subsequently, we underscore the processability and scalability of the Se-alloyed $Te–TeO_x$ semiconductor. The channel regions for the aforementioned TFT analysis were patterned using a metal shadow mask to mitigate the gate-leakage current and ensure reliable parameter extraction. We also found the compatibility of Se-alloyed $Te–TeO_x$

channel patterning with standard photolithography, highlighting its feasibility for industrial manufacturing. Further examination of the Se-alloyed $Te–TeO_x$ TFT array over a 10 cm (4 inch) wafer shows high device uniformity and reproducibility (Fig. 3f). The wafer-scale deposition of Se-alloyed $Te–TeO_x$ thin films costs around only US$0.3 for raw powder materials and takes just a few seconds, representing a cost-effective and high-throughput manufacturing process.

Finally, to demonstrate the compatibility of Se-alloyed $Te–TeO_x$ with established n-type metal–oxide technology, we integrated various complementary logic devices, including inverters, NAND gates and NOR gates. An inverter, incorporating n-channel $In_2O_3$ and p-channel Se-alloyed $Te–TeO_x$ TFTs, showed full-swing characteristics with rapid voltage transitions (Fig. 4a–c). A high voltage gain of 1,300 was obtained at a supply voltage ($V_{DD}$) of 20 V. The high gain is crucial for signal propagation and logic operations in circuits[38]. The inverter also delivered a high noise margin (82% of $V_{DD}$/2), indicating robust tolerance to noise and input signal variation for cascaded integrated circuit applications. The circuit leakage current as a function of $V_{DD}$ is shown in Extended Data Fig. 9. To enable a lower current level, future efforts could focus on reducing power supply, downsizing TFT and adjusting of the onset voltage of Se-alloyed $Te–TeO_x$ TFT to around 0 V. Two essential logic gates, NAND and NOR, were also constructed, delivering the correct logic function with an ideal rail-to-rail output voltage corresponding to their input states (Fig. 4d–f and Extended Data Figs. 10 and 11).

In conclusion, we have demonstrated high-performance stable p-channel TFTs using amorphous mixed-phase $Te–TeO_x$-based semiconductors through the scalable thermal evaporation method. The proposed Se-alloyed $Te–TeO_x$ shows superiority over reported emerging amorphous p-type semiconductors, showing outstanding electrical performance, cost-effectiveness, high stability, scalability and processability. The fabrication procedures align seamlessly with industry production lines and back-end-of-line technology. The hybrid-phase strategy introduces a new approach to designing new-generation stable amorphous p-type semiconductors. We expect this study can initiate research topics regarding semiconductor devices and promote the realization and commercialization of cost-effective, large-area, stable and flexible complementary electronic devices and circuits.

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

## Methods

### Thin-film fabrication and characterizations

$TeO_2$ (≥97%) and Se (99.99%) powders were purchased from Sigma-Aldrich and directly used as evaporation source. The $Te–TeO_x$ based films were deposited using a thermal evaporator placed in a $N_2$-filled glove box following a standard procedure to minimize the possible contaminations and vapour toxicity. The mixed $TeO_2$ (400 mg) and Se (12 mg) powders were loaded in a tungsten boat for optimal conditions. The substrate temperature was maintained at room temperature, and the vacuum pressure before evaporation was around $6 × 10^{-6}$ Torr. The distance between the $TeO_2$/Se powder-loaded tungsten boat and substrate holder was around 20 cm. The thickness of the Se-alloyed $Te–TeO_x$ films was monitored during deposition and the shutter was closed once the desired thickness was obtained. The evaporated samples were annealed at different temperatures for 30 min in ambient air. The crystal structures of the films were analysed using XRD with Cu Kα radiation (Bruker D8 ADVANCE). The HRTEM images and fast Fourier transform patterns were obtained using HRTEM (JEOL JEM 2100F). The X-ray photoelectron spectroscopy analysis was performed using a PHI 5000 VersaProbe instrument (Ulvac-PHI). The Se alloying percentages were characterized using high-resolution inductively coupled plasma mass spectrometry (Thermo Element XR) by dissolving the deposited films in a $HNO_3$ and HCl mixed acid solvent. Te and Se K-edge X-ray absorption spectra of the Se-alloyed $Te–TeO_x$ films were collected on the BL10C beam line at the Pohang light source (PLS-II) with top-up mode operation under a ring current of 250 mA at 3.0 GeV. Comprehensive measurement details and analysis of X-ray absorption spectroscopy are summarized in the Supplementary Methods (Supplementary Figs. 1–4 and Supplementary Tables 1 and 2).

### DFT calculation

The DFT calculations were performed using the Vienna ab initio simulation package code[39]. Projector augmented wave pseudopotentials[40,41] and a kinetic energy cut-off of 500 eV were used. The PBE exchange-correlation functional was used[42]. The amorphous structure was modelled as a cubic supercell using melt-and-quench molecular dynamics simulations. A random initial structure was melted at 3,000 K for 3 ps and then quenched to 0 K at a rate of $-1$ K $fs^{-1}$. The residual forces were relaxed to less than 0.01 eV/Å. A single $k$-point at (1/4, 1/4, 1/4) in the cubic Brillouin zone was used. The molecular dynamics time step was set to 1 fs. The supercell volume was fixed during the molecular dynamics simulations. We generated 50 different $TeO_{1.2}$ amorphous structures (samples) in 79-atom ($Te_{36}O_{43}$) cubic supercells, respectively, using melt-and-quench molecular dynamics simulations beginning with 50 different random initial structures. The calculated properties were the mean values obtained for the 50 samples. The supercell volume was fixed during the molecular dynamics simulations with a lattice constant of 11.6 Å in the cubic supercell.

### Device fabrication and characterization

A heavily doped Si wafer (resistivity 1–100 Ω cm) with a 100 nm thermally grown $SiO_2$ was used as the gate electrode and the dielectric layer. The Se-alloyed $Te–TeO_x$ channels were deposited on $SiO_2$ as channel layers identically using the aforementioned procedure, followed by the postannealing in ambient air at different temperatures for 30 min. The shadow mask was then covered on the substrate to obtain the patterned channel layers. Ni source and drain electrodes (40 nm) were deposited with a shadow mask using thermal evaporation to construct a bottom-gate top-contact TFT. The channel length and width were 250 and 1,000 μm, unless stated otherwise. All TFT electrical characterizations were characterized using a Keithley 4200 SCS at room temperature in a $N_2$ glove box. The $\mu_h$ was extracted by the McLarty technique in the linear regime. For logic gate integration, a 20 nm Ni was deposited as the patterned gate electrode with a 100 nm atomic-layer-deposited $HfO_2$ as the dielectric. To prepare the $In_2O_3$ solution, 0.1 M indium nitrate hydrate was dissolved in 2-methoxyethanol followed by stirring for 3 h. The precursor was spun at 5,000 rpm for 30 s followed by 250 °C annealing for 0.5 h. Thermal evaporated Al was deposited as the source and drain electrode.

## Data availability

The data supporting the findings of this study are included within the paper and its Supplementary Information.

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

**Acknowledgements** We acknowledge the insightful comments from H. Hosono (Tokyo Institute of Technology). A.L. acknowledges Y. Xu (Nanjing University of Posts and Telecommunications) for suggesting the reliable extraction of $\mu_h$ using the McLarty technique. This study was supported by the Ministry of Science, and ICT (information and communications technology) through the National Research Foundation, funded by the Korean Government (grant nos. RS-2023-00260608 and NRF-2021R1A2C3005401), the National Natural Science Foundation of China (grant no. 52372106), the BK21 FOUR program for Education Program for Innovative Chemical Engineering Leaders of the National Research Foundation of Korea (NRF) grant funded by the Korean government, and the Samsung Display Corporation.

**Author contributions** A.L., H.Z. and Y.-Y.N. conceived the study. A.L. performed the experiments and analysed the data. Y.-S.K. performed the DFT calculations. M.G.K. performed the XANES and EXAFS measurements and analysis. H.Z., S.B., Y.R., T.C. and T.Z. assisted with the characterization and discussion. A.L. and H.Z. designed and analysed the circuit. A.L., H.Z., Y.-S.K. and Y.-Y.N. wrote the manuscript. All the authors contributed to the final version of this manuscript.

**Competing interests** The authors declare no competing interests.

**Additional information**
**Correspondence and requests for materials** should be addressed to Ao Liu, Huihui Zhu or Yong-Young Noh.

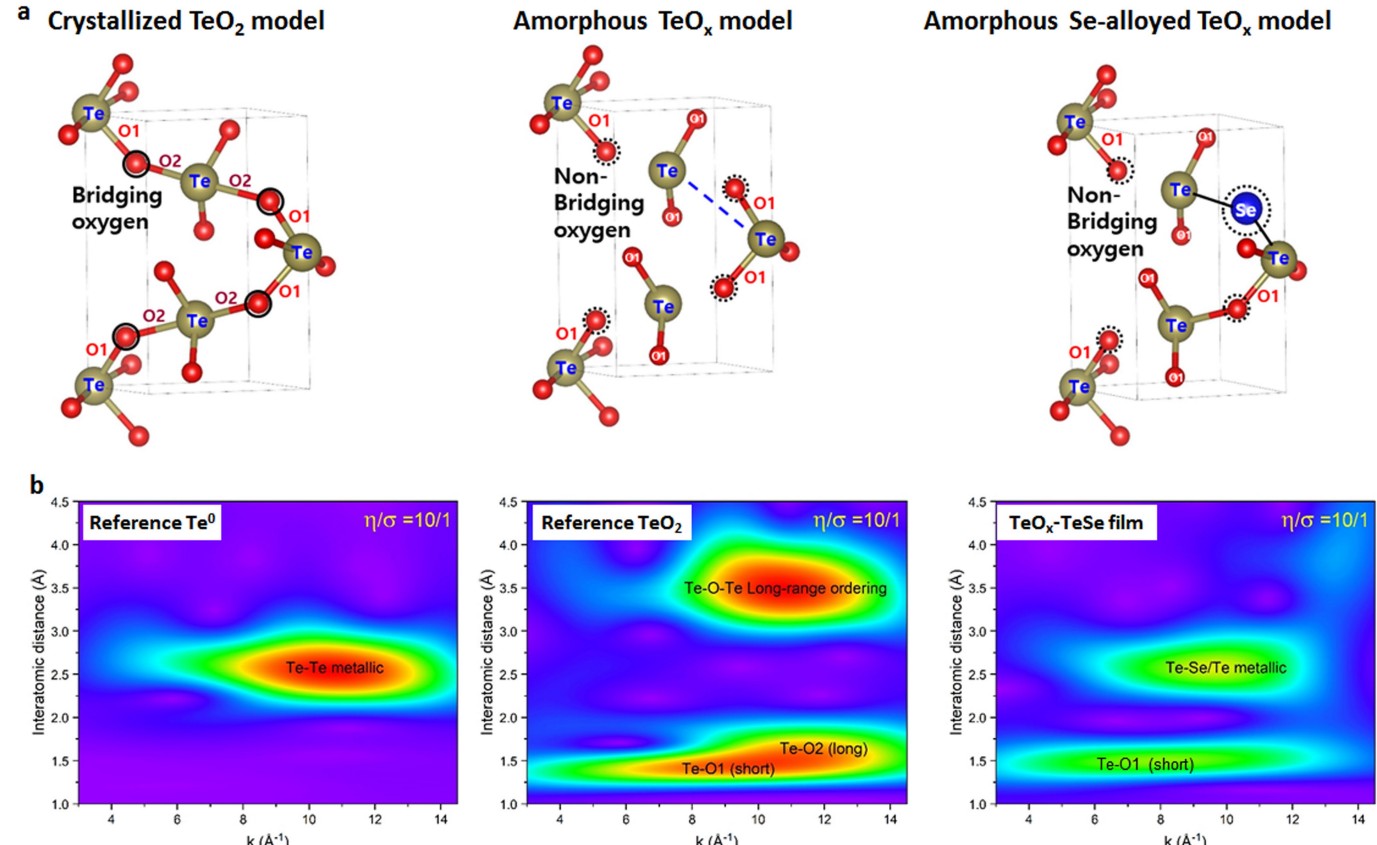

**Extended Data Fig. 1 | EXAFS analysis based on different TeO$_x$ models.**
**a.** Local bonding state of crystalized TeO$_2$, amorphous sub-stoichiometric Te-TeO$_x$ (longer Te-O2 bond partially broken with the formation of Te-Te metallic bonding), and amorphous Se-alloyed Te-TeO$_x$ (alloyed Se is located at oxygen vacant site with the formation of Te-Se-Te bonding). **b.** Wavelet-transformed $k$-$r$ space correlations for EXAFS spectra.

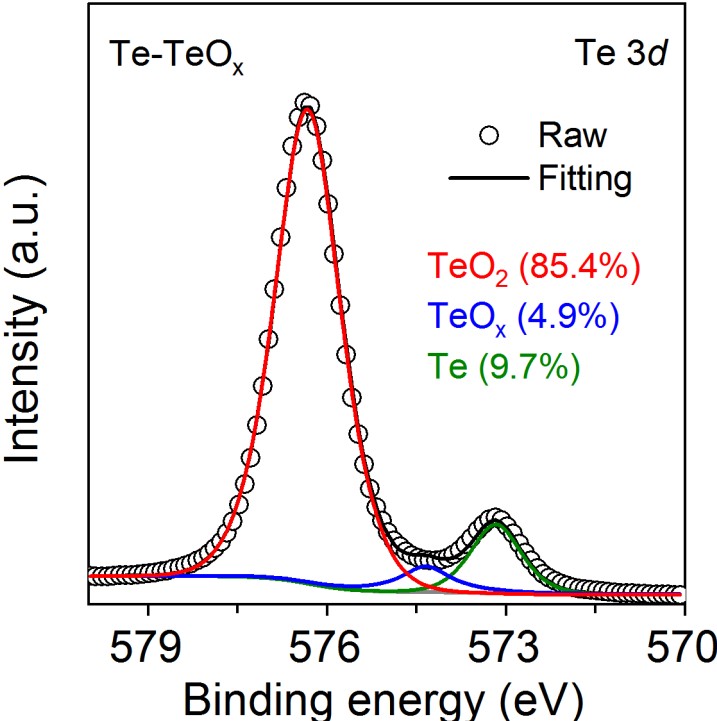

**Extended Data Fig. 2 | XPS Te 3d spectra of the deposited Te-TeO_x thin film.** The XPS measurement was surface scans without etching to avoid the possible influence on the local bonding environment by sputtering.

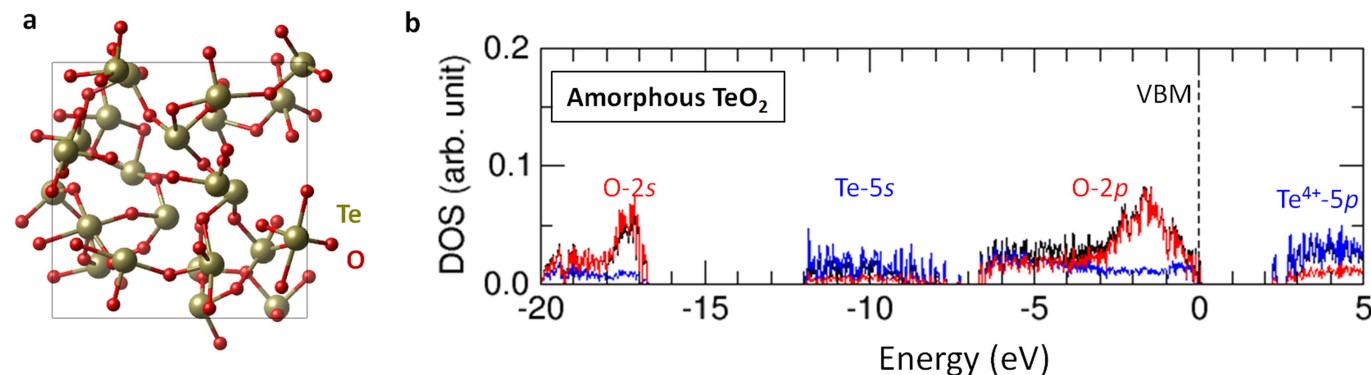

**Extended Data Fig. 3 | Atomic and electronic band structures of amorphous tellurium dioxide (TeO₂). a**. Atomic structure of amorphous TeO$_2$. **b**. Corresponding local density-of-states, showing the O 2$s$, Te 5$s$, and O 2$p$ filled (core or valence) states, and Te$^{4+}$ 5$p$ empty (conduction) states.

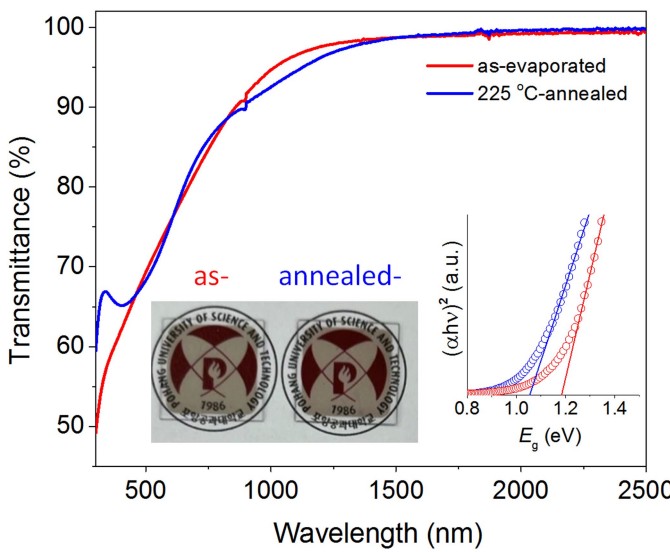

**Extended Data Fig. 4 | Optical analysis of the Te-TeO$_x$ thin films.** Optical spectra, bandgap extraction, and optical pictures of as-evaporated and 225 °C-annealed Te-TeO$_x$ thin films on glass. The pure TeO$_2$ was reported to have a wide $E_g$ of ~3.5 eV. In this study, the resulting film consists of a mixed phase of sub-stoichiometric TeO$_x$ and metallic Te. Because of the narrow $E_g$ of Te (~0.35 eV), the final film shows a smaller $E_g$ compared to that of the pure TeO$_2$.

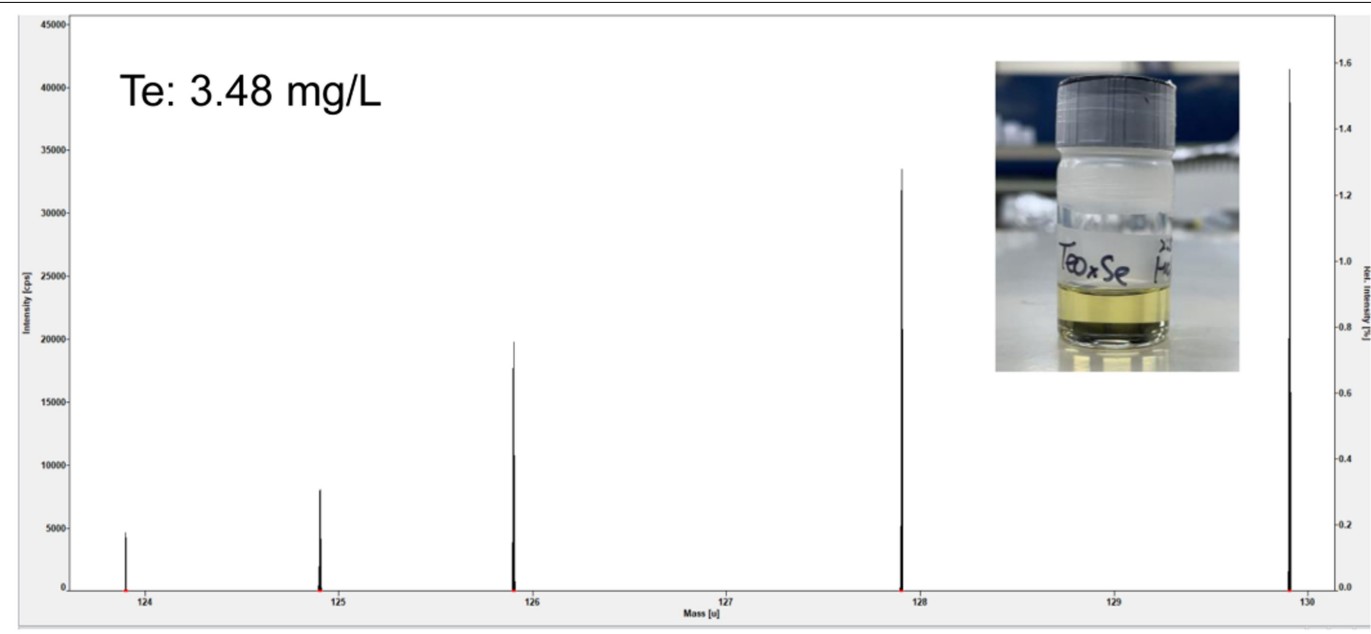

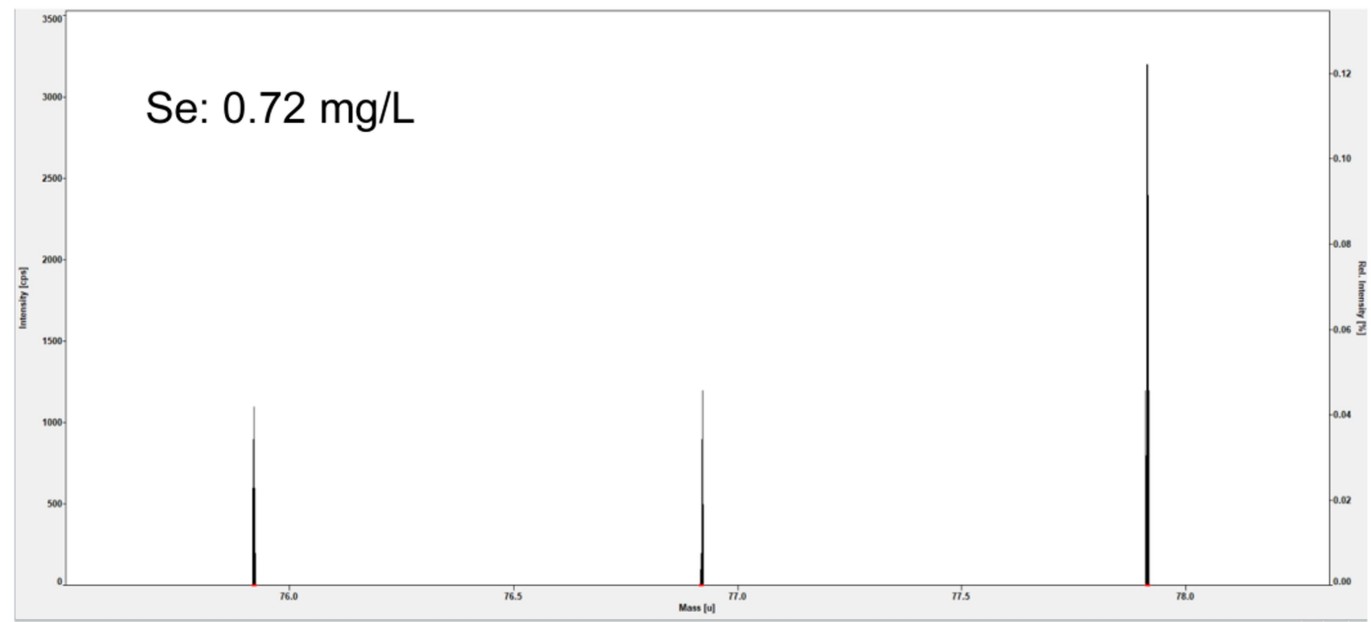

**Extended Data Fig. 5 | High-resolution inductively coupled plasma mass spectrometry spectra of Te and Se characterization elements.** The inset picture shows the solution used, which is prepared by dissolving the deposited Se-alloyed $TeO_x$ thin film in mixed $HNO_3$ and HCl acid.

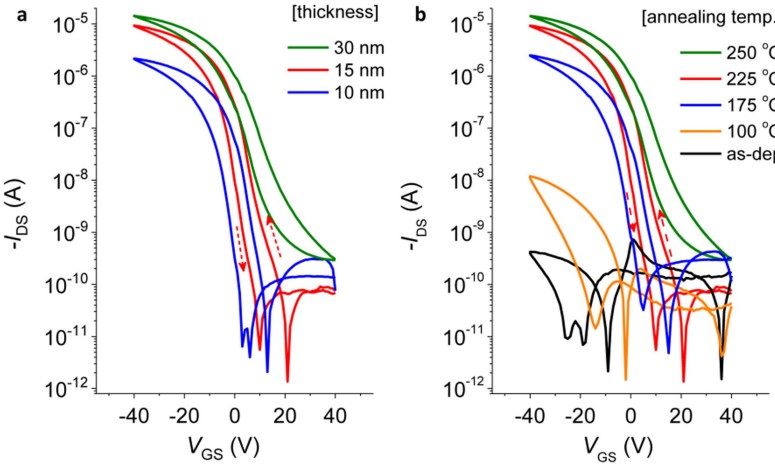

**Extended Data Fig. 6 | Transfer characteristics of Se-alloyed Te-TeO$_x$ TFTs with different conditions. a**. Transfer curves of Se-alloyed Te-TeO$_x$ TFTs fabricated with different channel thicknesses (channel annealing temperature is 225 °C). **b**. Transfer curves of Se-alloyed Te-TeO$_x$ TFTs deposited with different channel layer postannealing temperatures. All the hysteresis direction is counterclockwise.

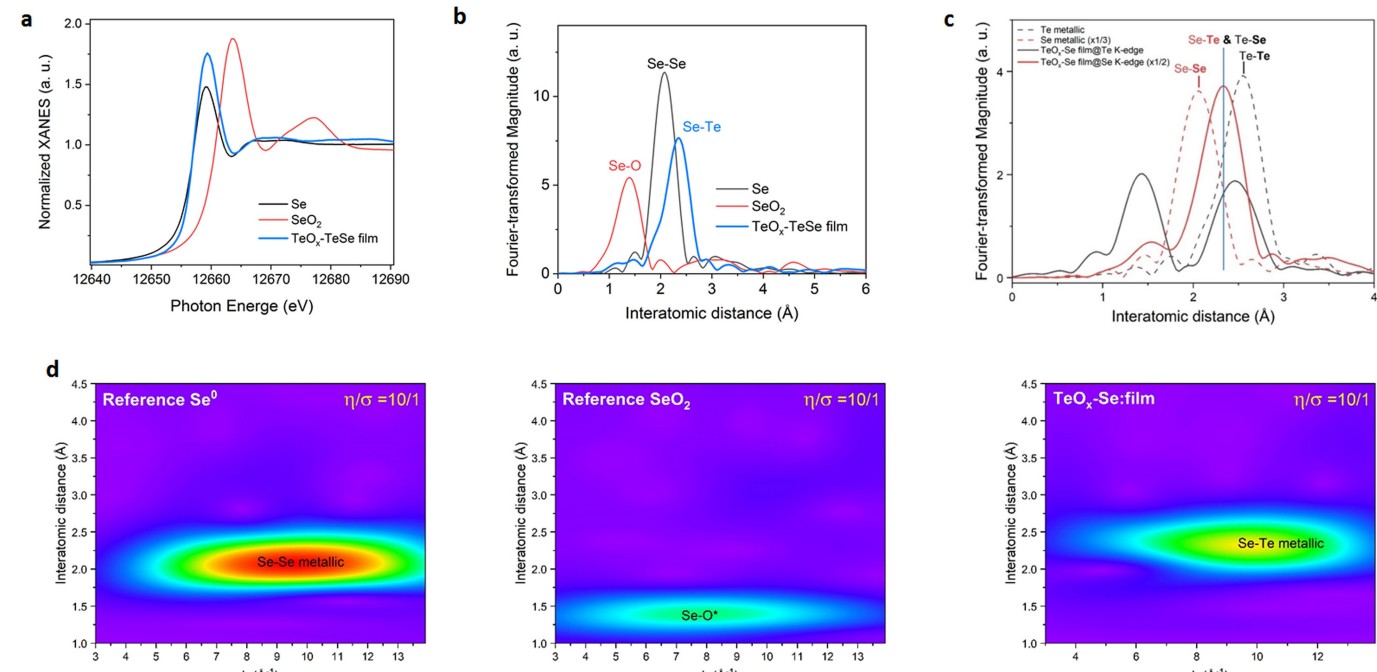

**Extended Data Fig. 7 | XANES/EXAFS analysis of on different Te-TeO$_x$ based thin films. a**. Normalized XANES spectra of the Se-alloyed Te-TeO$_x$ (TeO$_x$-TeSe) film at Se K-edge, compared to reference materials of Se and SeO$_2$. **b**. Corresponding Fourier transform (FT) of Se K-edge $k^3$-weighted EXAFS spectra. **c**. Comparison of Fourier-transformed features for Te K-edge (black solid line) and Se K-edge (red solid line) EXAFS spectra, compared to those of reference metallic Te and Se powder (dashed line). **d**. Corresponding Wavelet-transformed $k$-$r$ space correlations for EXAFS spectra of Se, SeO$_2$, and TeO$_x$-TeSe film, respectively.

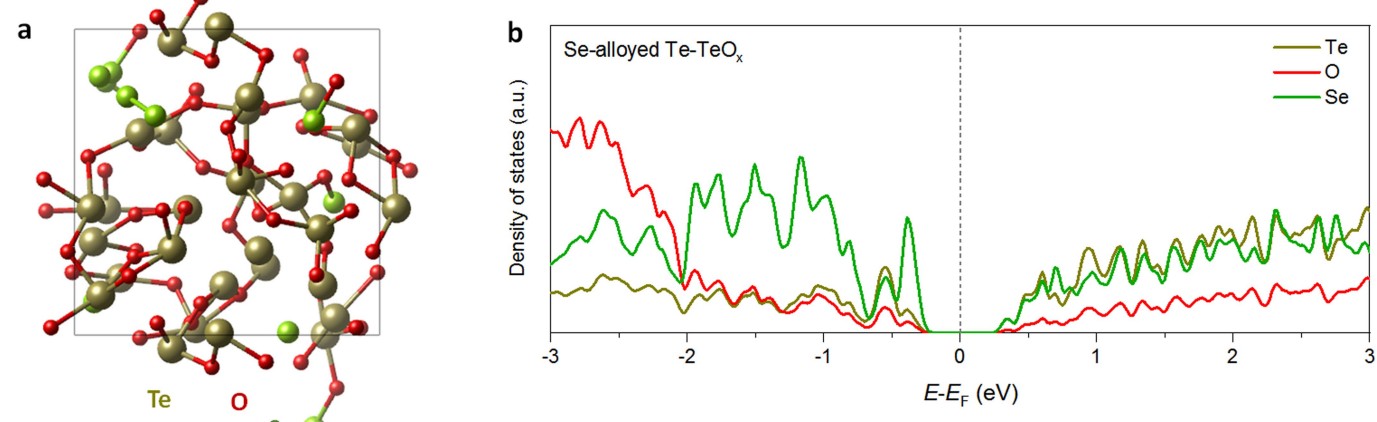

**Extended Data Fig. 8 | Atomic and electronic band structures of the Se-alloyed Te-TeO$_x$. a**. Atomic structure of amorphous Se-alloyed Te-TeO$_x$ generated in DFT (Te:O atomic ratio = 1:1.2; Te:Se atomic ratio = 3:1, film density = 5.6 g/cm$^3$). **b**. Projected DOS of the O 2$p$ (red), Te 5$p$ (dark yellow), and Se 4$p$ (green) states in DFT-PBE.

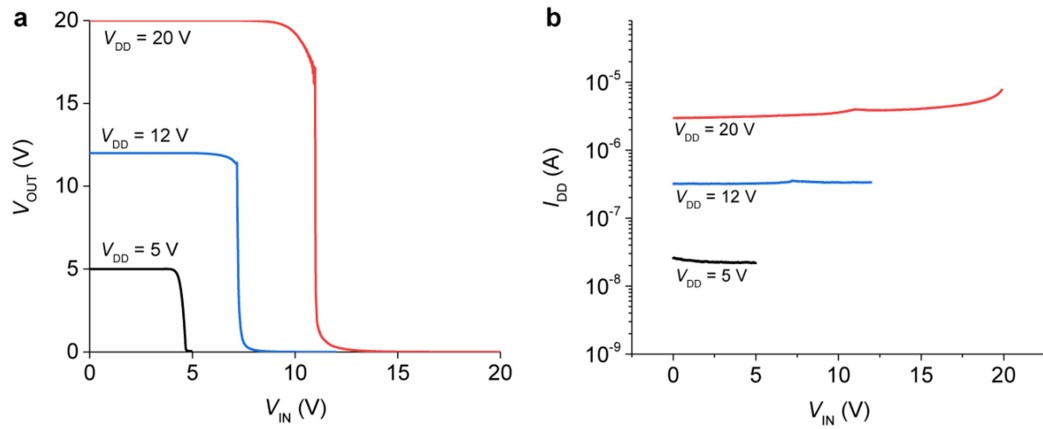

**Extended Data Fig. 9 | CMOS inverter characterisations. a**. Voltage transfer curves of one complementary inverter based on n-channel $In_2O_3$ and p-channel Se-alloyed Te-TeO$_x$ TFTs at different $V_{DD}$. **b**. Corresponding circuit leakage current.

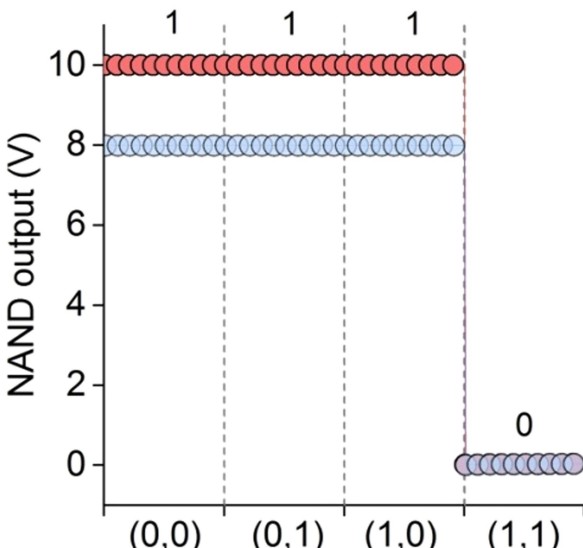

**Extended Data Fig. 10 | CMOS NAND.** The rail-to-rail NAND gate with the working voltage of 10 and 8 V, respectively.

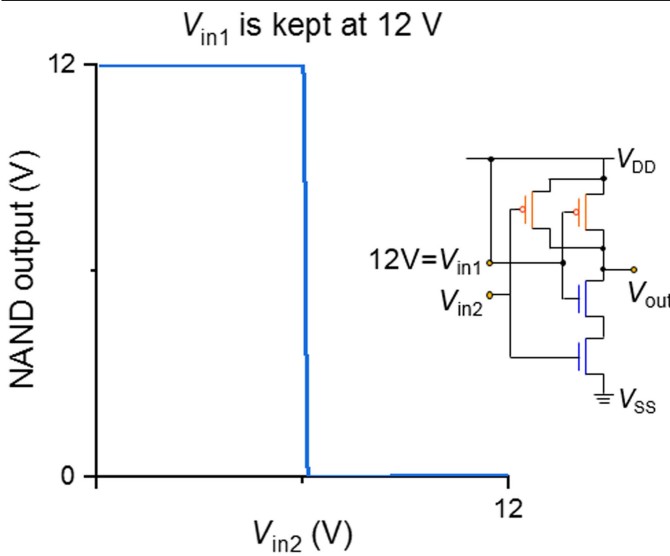

**Extended Data Fig. 11 | CMOS NAND with fixed $V_{in1}$.** Plotting of the NAND output *vs* input $V_{in2}$ while $V_{in1}$ was kept at a $V_{DD} = 12$ V.

**Extended Data Table 1 | TFT parameter comparison**

| Method | Channel material | Annealing temp. (°C) | $\mu_h$ (cm$^2$ V$^{-1}$ s$^{-1}$) | $I_{on}/I_{off}$ | Stability | Ref. |
|---|---|---|---|---|---|---|
| Chemical vapor deposition | a-Si:H | 250-350 | ≤0.1 | ~$10^6$ | good | 43 |
| Solution process/thermal evaporation | Organic | 100-150 | <0.1 | ~$10^5$ | poor | 44 |
| Pulsed laser deposition | black phosphorus | 150 | 14 | ~$10^2$ | poor | 45 |
| Spin coating | CuSnSGaO | 250 | 1.75 | ~$10^4$ | -- | 46 |
| Spin coating | CuI | 100 | -- | ~$10^4$ | poor | 47 |
| Spin coating | NiO | 250 | 0.48 | ~$10^3$ | -- | 48 |
| Sputtering | CuCrO$_2$ | 150 | 0.3 | ~$10^4$ | -- | 49 |
| Thermal evaporation | SnO | 300 | 5.59 | 40 | -- | 50 |
| Pulsed laser deposition | CuNiSnO | 100 | 1.37 | ~$10^5$ | -- | 51 |
| Magnetron sputtering | ZnRhCuO | RT | 0.079 | ~$10^3$ | -- | 52 |
| Thermal evaporation | Se-alloyed Te-TeO$_x$ | 225 | ~15 | ~$10^7$ | good | this work |

Representative parameters of p-channel TFTs based on different amorphous p-type semiconductors ("--" means no mention in the reported literature). Refs. 43–52.