## [Peer Review File · Nature]

Manuscript Title: Selenium alloyed tellurium oxide for amorphous p-channel transistors

Reviewer Comments & Author Rebuttals

Reviewer Reports on the Initial Version:

Referees' comments:

Referee #1 (Remarks to the Author):

The manuscript reports high-performance p-channel amorphous TeO_x-TFTs and the related CMOS inverter and demonstrates significant progress of p-channel oxide-TFTs. However, the main concern remains channel material, which should be completely resolved for publication.

1. The largest concern is that the channel is possibly a-Te. Several previous studies show that metal Te easily to be amorphous form by oxygen and Se impurity. [e.g. Taiky Kim et al, High-Performance Hexagonal Tellurium Thin-Film Transistor Using Tellurium Oxide as a Crystallization Retarder, IEEE Electron Device Letters (Year: 2023 | Volume: 44, Issue: 2) Moreover, the TFT device performances of the presented device are very similar as the previously reported p-channel metal Te-TFTs including oxygen impurity. Basically, these channels are conductive and operate in depletion mode. Therefore, more deep material characterization such as local-atomic structure analysis using EXASF is strongly required to eliminate amorphous metal Te channel. The author already provided the surface-sensitive XPS in the SI file, but the data doesn't guarantee that the channel is TeO_x channel.
2. Amorphous model used for the DFT calculation seems too speculative. The presented model should be verified by local atomic analysis and film density analysis.
3. The material properties such as band gap and Hall-effect mobility should be provided. The band gap information is strongly required.
4. Se concentration should be measured with good accuracy. Since the actual detection limit of XPS is not good and in general, it corresponds to several % if the peak appears. Therefore, the authors should double-check the Se concentration by using other chemical analysis.
5. The authors should provide a statistical analysis of the inverter performances such as gain and noise margin. The authors should discuss the reasons if there are devices with low performances.
6. The reviewer thinks that the authors should provide a detailed fabrication process to reproduce the same device performances for others because there are many papers with poor reproductivity in this field. Especially, in high-impact journals. For example, the uniform 2-nm channel using a conventional thermal evaporator with a powder source seems very hard to reproduce. The Se doping in thermal

evaporation sounds not easy to repeat the same data.

Referee #2 (Remarks to the Author):

This manuscript described the synthesis of a non-stoichiometric tellurium oxide with selenium doping which forms a p-type semiconductor. The authors characterize some of its material properties and demonstrate its use in thin film transistors, inverters and NAND/NOR logic gates when combined with n-type In₂O₃. The authors also perform DFT to provide physical insight. The work is certainly of interest to the Nature Electronics community, and the experimental work appears to be done correctly and reported clearly. However there are several issues that should be addressed in order for this manuscript to be considered for publication.

The manuscript lacks citations of existing theoretical and experimental work on TeO₂ as an electronic material. A good place to start would be DOI: 10.1039/C8NR01028E, and the references therein. The authors should explain in the manuscript, with appropriate citations, what is already known in the field about this material, in order for the reviewer and reader to understand: (a) why they have chosen to pursue this material, (b) how their results differ from those previously obtained by others, and thereby (c) the authors' contribution to the field.

The theory work appears insufficient to support the claims in the manuscript in the top half of page 3. Specifically, statements about acceptor-like states need to be supported by calculations of a variety of intrinsic defects to determine their energy levels and likelihood of formation. If this is not possible, these claims should be amended to be made more limited in scope, or removed. In addition, the convergence of the results for the given supercell size should be given, and the theoretical bandgap and density of the amorphous TeO_x should be compared to that obtained from experiment.

The meaning of the sentence "In fact, V_o is an identical species with TeO and Te²⁺" is unclear. How is an oxygen vacancy identical to neutral or ionized metals in the metal-oxide matrix? Please rewrite this sentence to clarify the meaning.

Was any hysteresis observed in device I-V sweeps? Hysteresis appears to be visible in Fig 3b. Please show forward and reverse sweeps in Fig 3a, Supplementary Figs. 4, 5 and 7, and comment on hysteresis or self-heating effects observed.

In the second sentence of the introduction, and again later on page 3 (line 70, line 81), the authors dismiss p-type oxides, with erroneous blanket statements about their VBM composition being made of O 2p orbital. Mention here should be made of various Cu and Sn oxides, which have VBM that are hybridized (Cu 3d and O 2p orbitals; Sn 5s and O 2p), leading to measured hole mobility of > 100

cm²/V/s in Cu₂O bulk crystals and predicted similarly high hole mobility of stannate oxides, even though thin films have not yet been shown to obtain the same properties. These are interesting materials and active research areas, which compete with the technology demonstrated in this manuscript. See, for example: DOI 10.1021/acs.chemmater.6b03306, 10.1063/1.3231869, <https://doi.org/10.1063/5.0078548>, and 10.1021/acs.chemmater.0c03495

In Fig. 4, please indicate in the caption the meaning of the red and blue boxes in part (d).

Please specify in the figure caption or Methods the W/L of the NMOS and PMOS devices used in the inverter, NAND and NOR gates.

For Supplementary Fig 2, please specify the etch time or depth for the XPS traces (i.e. are they surface scans or taken after some sputtering has occurred?)

Tellurium and selenium are well known to be acutely toxic. In order for the work to be repeatable and for the reader to understand possible sources of contamination/impurities, please specify in the Methods sections the safety precautions taken when, for example, loading the materials into the evaporation crucible.

The scale bar in Fig. 1d needs to be enlarged.

Referee #3 (Remarks to the Author):

The authors bring an interesting study towards amorphous p-type tellurium oxide transistors, which is an important research topic to complement the n-type IGZO transistor for circuit and backplane applications. However, nowadays, complementary circuits are being commercialized, combining IGZO and p-type LTPS transistors, but I would agree to the authors that it would have a clear advantage if a p-type oxide can be found which enables a more simplified process flow. I would recommend to add in the introduction a paragraph on the most promising options to complement the n-type metal-oxide transistors such as p-type LTPS.

Despite the fact that it is an interesting topic, from circuit and application perspective, there are several key items missing, of which some of them are listed below.

(1) The characteristics at Fig. 3 show an onset voltage at about +20V VGS, or a positive threshold voltage. This implies that at 0V VGS, there is still substantial current flowing $>10^{-8}$ amps. A complementary circuit will therefore still have considerable leakage current, which is not solving the issue of p-type leakages. A method to control the threshold voltage (onset voltage) at 0V would be very beneficial, to drop the leakage current with a few orders of magnitude, and therefore increase the on/off ratio for a digital circuit.

(2) Applying 40V on a 100nm thermally grown SiO₂ is of course feasible, however, the field is very large.

This may help to push the mobility if it is dependent on the field. Is this a field-dependent mobility, or is it stable at all fields? Moreover, for a typical application, supply voltages should be kept small, e.g. below 10V or 5V. The real on/off ratio of the circuit will therefore degrade. The inverters and logic gates are shown at 20V, however the community and readers of this work would be interested how much it scales with downscaling supply voltage.

(3) Another important and missing figure with respect to the oxide TFT circuit are the currents, as indicated in my previous comments. It would be necessary to include in Fig. 4b the supply currents of the inverter. This will be a key figure of merit to qualify the work and proper functioning of the CMOS inverter. The 'real' static leakage current can be calculated from those figures, instead of starting from Fig. 3, which is not indicative.

(4) The circuits have been based on the combination of the studied TeOx:Se and InOx transistors. There is only a link to a publication with regards to the InOx transistors, however, in order to evaluate reproducibility of this work, characteristics of the n-type device would be useful and interesting for the reader. Moreover, the behavior of the InOx transistors and its on/off ratio are also important for the behavior of the inverter, especially characterized at the dielectric of the inverter/logic gates.

(5) The channel lengths of the devices are pretty large and should be indicated in the figures. A 200 μm channel length is actually not on par with requirements of applications, as those require channel lengths close to 1 μm . In addition, a 200 μm channel length results in a large output resistance, which enables a large gain device! Therefore, this figure is a bit misleading. It would be interesting to see the scalability of this parameter at smaller channel lengths.

(6) Fig. 4 e and f may prove that the NAND/NOR gate is operating correctly, however, the way it is presented can be lead to misinterpretation, especially with respect to gain. A better method would be to plot all different characterized curves on top of each other, namely:

- input A = 0 and input B varies from 0V-12V.
- Input B = 0 and input A moves from 0V-12V.
- Input A and B move together from 0V – 12V.

As such, the gain, the switching threshold, the currents and variations with respect to input states will become visible.

(7) In the section device fabrication, it is written that 100-nm thermally grown SiO₂ was used. However, the TFTs for the logic gates have been made on HfO₂ as gate dielectric. Please make this clear in the text and in the (caption of the) figures. Applying 40V on this high-k dielectric implies a very large field. The same comment holds for comment (2): how does it scale down, as HfO₂ should enable much lower supply voltage operation, closer to the specifications of the applications. In addition, the characteristics of both TFTs are missing when characterized on HfO₂, which may have a large impact on V_t, on/off ratios and mobilities.

(8) Fig. 3 indicates PBS and NBS with different time durations. Do the authors have an idea about the

impact of light on the stability of the devices?

Referee #4 (Remarks to the Author):

The paper describes material properties, density functional calculations and thin film transistor (TFT) characteristics of amorphous evaporated Se-doped TeO_x. The authors demonstrate p-type TFTs with low threshold voltage, high mobility, reasonable stability, manufacturability and complementary circuit integration. There has been a long search for a p-type TFT material that matches the properties of the amorphous n-type metal oxide TFTs, but previous examples have lacked some major aspect of the required properties. The results presented here are therefore interesting. The subthreshold properties are less than ideal and so is the stability but future work may improve these.

Statistical mobility data is reported for TFTs from one wafer. The paper should also provide information about wafer-to-wafer statistics.

The statement that V_0 is identical to T_0 and Te_{2+} (page 4 near bottom) needs some further explanation. The bias stress instability is attributed to charge trapping rather than defect creation. Does this mean charge trapping in the dielectric, at the interface or in the TeO_x channel, and why?

The authors claim to have demonstrated manufacturability, but the devices they make are on Si/SiO₂ wafers, rather than using large area technology with a deposited dielectric. While their approach is sufficient for a first publication, any further information they can give about the properties of their TFTs on a deposited oxide would be very valuable.

There is a very recent paper by Choi et al (Electron Device Letters 44, 269, 2023) which describes an alternative TeO TFT made by reactive sputtering. The authors may not have been aware of the paper when they submitted but should include this reference.

The paper is an important addition to TFT technology and the device has a reasonable probability of becoming an industry standard. The paper should be accepted after the authors have responded to the points made above.

Author Rebuttals to Initial Comments:

Reviewer #1: The manuscript reports high-performance p-channel amorphous TeO_x -TFTs and the related CMOS inverter and demonstrates significant progress of p-channel oxide-TFTs. However, the main concern remains channel material, which should be completely resolved for publication.

Our reply to Reviewer 1:

We sincerely thank the reviewer's important comments and suggestions. Here below is our point-by-point response. The manuscript was revised accordingly as highlighted.

RIQ1. The largest concern is that the channel is possibly a-Te. Several previous studies show that metal Te easily to be amorphous form by oxygen and Se impurity. [e.g. Taiky Kim et al, High-Performance Hexagonal Tellurium Thin-Film Transistor Using Tellurium Oxide as a Crystallization Retarder, IEEE Electron Device Letters (Year: 2023 | Volume: 44, Issue: 2)] Moreover, the TFT device performances of the presented device are very similar as the previously reported p-channel metal Te-TFTs including oxygen impurity. Basically, these channels are conductive and operate in depletion mode. Therefore, more deep material characterization such as local-atomic structure analysis using EXAFS is strongly required to eliminate amorphous metal Te channel. The author already provided the surface-sensitive XPS in the SI file, but the data doesn't guarantee that the channel is TeO_x channel.

Our reply: We thank the reviewer's insightful comments. Following the advice, we conducted the Te K-edge XANES and EXAFS analysis (shown below). The Te K-edge XANES spectrum of TeO_x :Se bulk film is similar to a spectral feature of the TeO_2 phase, with oxide ratio of $\sim 60\%$ from a linear combination of XANES spectra for the reference Te and TeO_2 . The analysis also helps better understand the amorphous phase formation. Compared to reference TeO_2 powder with peak feature of shorter Te-O_1 and longer Te-O_2 bonds, the radial distribution function (RDF) of EXAFS for the deposited TeO_x :Se film shows the slight decrease of shorter Te-O_1 bond and the disappearance of both longer Te-O_2 and Te-O-Te long-range ordering. The generation of oxygen vacancy breaks the bridged bond of Te-O_2 and the undercoordinated Te leads to the loss of Te-O-Te long-range ordering, forming the amorphous structure.

The corresponding XANES/EXAFS data and statement were added in the revised manuscript (Fig. 1 and Supplementary Fig. 1).

Figure R1 (Figure 1d and 1e). (a) Te K-edge XANES spectra of TeO_x:Se film (~70 nm) and those of reference materials. (b) Corresponding Fourier-transformed radial distribution functions of Te K-edge k^3 -weighted EXAFS spectra.

For the given reference paper (cited as ref. 29), the idea/methodology/outcome are fundamentally different from our work.

1) The reference paper investigates oxygen ratio (P_o) during Te sputtering to form TeO₂ and reduce the random polycrystallinity in the Te film. The optimized TFT channel is highly crystallized with ~90% elemental Te (7% P_o with encapsulation, XRD in Fig. 1b). In this case, the result is still related to polycrystalline Te TFTs, which were widely reported with high hole mobilities (npj 2D Materials and Applications 6, 17, 2022). When the polycrystalline channel contains ~50% TeO₂ (33% P_o), the TFT hole mobility is greatly reduced to 3.6 cm²/Vs. One key challenge for Te TFTs is the high OFF current due to the narrow bandgap of Te ($E_g \sim 0.3$ eV). To enable good I_{on}/I_{off} in the reference paper, Al₂O₃ encapsulation was required to improve the current modulation, which was reported by the same group (npj 2D Materials and Applications 6, 4, 2022).

2) Our study used the evaporation method with TeO₂ powder to directly deposit TeO_x-based devices and addressed the long challenge for the high-performance amorphous p-channel oxide TFTs. Meanwhile, we rationalized why the amorphous films show good p-type character and explored the key role of Se doping on performance improvement (Se alloying was demonstrated to significantly degrade the Te TFT performance, Adv. Mater. 32, 2001329, 2020). To obtain enhancement-mode TFTs, further technical optimizations, for example, on the deposition approach, doping method/material, dielectric, and device engineering could be helpful.

Overall, the development of good p-type oxide TFT has been struggling in past decades, let alone the amorphous ones. New materials/design concepts have been highly needed. Even as an initial attempt, we believe this study would provide important inspiration and great confidence to the community to develop amorphous p-channel oxide TFTs for combination with n-type counterparts.

RIQ2. Amorphous model used for the DFT calculation seems too speculative. The presented model should be verified by local atomic analysis and film density analysis.

Our reply: Following the reviewer's good advice, we performed the XRR measurement (below figure) and the film density is ~5.6 g/cm³. In the original DFT calculation, a smaller 5 g/cm³ density was adopted for the amorphous model. The EXAFS analysis clarifies the Te:O atomic ratio in the bulk film is 1:1.2. In the revised manuscript (Figure 2), we updated the DFT model with newly measured experimental data. The corresponding statements were also revised (page 3) and the main explanation/conclusion is consistent.

Figure R2. The XRR spectrum and the simulation curve of the TeO_x-based thin film.

R1Q3. The material properties such as band gap and Hall-effect mobility should be provided. The band gap information is strongly required.

Our reply: We thank the reviewer's suggestion. The Hall mobilities of pristine TeO_x and optimized TeO_x:Se thin films are 24 and 67 cm² V⁻¹ s⁻¹ (page 5). The film band gaps were measured to be ~1.1 eV and added in the revised manuscript (page 3).

The stoichiometric TeO₂ possesses a wide E_g of ~3.5 eV. In this study, the smaller E_g could be related to the narrow bandgap Te ($E_g \sim 0.3$ eV) generation. Further explorations on the deposition procedure and the component modulation with alternative film growth methods like sputtering are expected to enable wider adjustment of optical properties for the potential demand for transparency.

Figure R3 (Supplementary Fig. 5). Optical spectra, bandgap extraction, and optical pictures of as-evaporated and 225 °C-annealed TeO_x-based thin films on glass.

R1Q4. Se concentration should be measured with good accuracy. Since the actual detection limit of XPS is not good and in general, it corresponds to several % if the peak appears. Therefore, the authors should double-check the Se concentration by using other chemical analysis.

Our reply: We thank the reviewer's comment. We measured the film XPS depth profile and the Se signal is stable and uniform over the whole film (Figure S2). We also consulted the XPS manager and confirmed the equipment is capable to measure atomic percentages up to a hundredths decimal point.

Figure R4 (Supplementary Fig. 2). XPS depth profile elemental distribution of as-evaporated and 225 °C-annealed TeO_x:Se thin films.

R1Q5. The authors should provide a statistical analysis of the inverter performances such as gain and noise margin. The authors should discuss the reasons if there are devices with low performances.

Our reply: We thank the reviewer’s suggestion. In this study, the given gain and noise margin values are representative data. We tested three batches of inverters, which all delivered a high gain of over 1000 with a noise margin of ~80%. The result is reasonable because the thermal evaporation gives high uniformity and reproducibility of p-type oxide films, and the fabrication of n-type oxides is highly mature as well. Furthermore, both are stable with adjustable/comparable electrical performance, yielding highly-reproducible inverter performance.

R1Q6. The reviewer thinks that the authors should provide a detailed fabrication process to reproduce the same device performances for others because there are many papers with poor reproductivity in this field. Especially, in high-impact journals. For example, the uniform 2-nm channel using a conventional thermal evaporator with a powder source seems very hard to reproduce. The Se doping in thermal evaporation sounds not easy to repeat the same data.

Our reply: We thank the reviewer’s good suggestion. For conventional p-type metal oxide TFTs, the deposition window is typically narrow. By using thermal evaporation, the deposition process is easily reproduced by loading mixed TeO₂ and Se powders in the tungsten boat. For easy reference, more details on TeO₂/Se powder weighing and evaporation program were added in the Methods.

We agree with the reviewer that 2-nm channel deposition may raise concerns for connectivity/uniformity. In this study, the optimized TeO_x-based channel thickness is ~6 nm, which delivers uniform morphology and should be easily reproducible. Given that the remaining data can still clearly show the channel thickness-dependent TFT behavior, we decided to remove the 2-nm data from the Supplementary Figure 7 to avoid potential misleading/misunderstanding.

Our revision (page 7, Methods): The mixed TeO₂ (220 mg) and Se (12 mg) powders are loaded in a tungsten boat for the optimal condition. The evaporation program includes the bulk intensity of 19.3, Z-factor of 0.38 and tooling factor of 83.5%.

Reviewer #2: This manuscript described the synthesis of a non-stoichiometric tellurium oxide with selenium doping which forms a p-type semiconductor. The authors characterize some of its material properties and demonstrate its use in thin film transistors, inverters and NAND/NOR logic gates when combined with n-type In_2O_3 . The authors also perform DFT to provide physical insight. The work is certainly of interest to the Nature Electronics community, and the experimental work appears to be done correctly and reported clearly. However there are several issues that should be addressed in order for this manuscript to be considered for publication.

Our reply to Reviewer 2:

We sincerely thank the reviewer's important comments and suggestions. Here below is our point-by-point response. The manuscript was revised accordingly as highlighted.

R2Q1. The manuscript lacks citations of existing theoretical and experimental work on TeO_2 as an electronic material. A good place to start would be DOI: 10.1039/C8NR01028E, and the references therein. The authors should explain in the manuscript, with appropriate citations, what is already known in the field about this material, in order for the reviewer and reader to understand: (a) why they have chosen to pursue this material, (b) how their results differ from those previously obtained by others, and thereby (c) the authors' contribution to the field.

Our reply: We thank the reviewer's good suggestion. We searched the literature and found three related references. They all focus on the orthorhombic crystalline $\beta\text{-TeO}_2$, which was predicted with good p-type property (cited as ref. 31-33). By contrast, our work explores the amorphous nonstoichiometric TeO_x -based compound, which has not been explored before.

At our initial attempt, we planned to evaporate TeO_2 powder to get $\beta\text{-TeO}_2$. Surprisingly, we found the deposited films were amorphous but with good p-type properties. However, for the amorphous stoichiometric TeO_2 , our DFT results show the VBM is mainly composed of localised O $2p$ states, indicating poor p-type character (Supplementary Figure 4). To clarify the good p-type property in the obtained amorphous composites, we conducted a series of analyses. Interestingly, we found the existence of certain amounts of undercoordinated tellurium, which play key roles for shallow acceptor formation and VBM dispersion. We also explored the key role of Se doping on the electrical performance improvement, which has not been reported for the $\beta\text{-TeO}_2$ either. Overall, the discovery in this work is novel, unexpected and fundamentally different from previous $\beta\text{-TeO}_2$ studies, representing a breakthrough in the amorphous p-channel TFT research field.

We added important information related to $\beta\text{-TeO}_2$ in the revised manuscript for readers' reference. The authors' contribution to the field (question (c)) was included in the Abstract/Conclusion parts.

Our revision (page 3): "This stands in marked contrast to previous studies on the orthorhombic low-dimensional $\beta\text{-TeO}_2$ with good p-type properties."

R2Q2. The theory work appears insufficient to support the claims in the manuscript in the top half of page 3. Specifically, statements about acceptor-like states need to be supported by calculations of a variety of intrinsic defects to determine their energy levels and likelihood of formation. If this is not possible, these claims should be amended to be made more limited in scope, or removed. In addition, the convergence of

the results for the given supercell size should be given, and the theoretical bandgap and density of the amorphous TeO_x should be compared to that obtained from experiment.

Our reply: We thank the reviewer's important comments. To find a shallow acceptor defect, all kinds of intrinsic defects such as vacancies, interstitials, anti-sites, and complexes should be considered and their formation energies and transition levels should be thoroughly investigated. This approach works well for a crystal system. In an amorphous system, the host atomic structure is not definite, and vacancy, interstitial, and anti-site defects become ambiguous. For example, O-vacancy and Te-interstitial are indistinguishable in amorphous TeO_x. Thus, we used another approach. We generate 50 different amorphous structures and look deeply into the atomic and electronic structures. Then, we can find a frequently found defect in the amorphous system. The Te-5*p* defect bands inside the bandgap in TeO_x are the common feature in all our 50 generated amorphous TeO_x, and 5 generated structures exhibit the shallow acceptor states. We modify this part in the revised manuscript with more details in the Method.

The theoretically calculated bandgap is ~0.91 eV, which is slightly lower than the experimental result of ~1.1 eV (Supplementary Fig. 5). Since DFT is known to underestimate the band gap, the discrepancy is reasonable. We performed XRR measurement to check the film density (~5.6 g/cm³), which is close to the adopted value (~5 g/cm³) for the original DFT calculation. We also conducted EXAFS measurement to perform the local atomic analysis (added as new Figs. 1d and 1e). In the revised manuscript, we updated the DFT model based on the newly measured experimental data and revised the corresponding statements (Figure 2). The main explanation/conclusion remains consistent.

To check supercell convergence in our DFT calculations, we generated 50 different TeO_{1.2} amorphous structures (samples) in 79-atom (Te₃₆O₄₃) cubic supercells, and calculated Te-O radial pair distribution functions (RPDF). Overall, the updated DFT model is well consistent with the EXAFS analysis and other characterizations, suggesting the amorphous structure/model generated by DFT are reasonable. For different supercell sizes, the calculated properties are not varied much, as listed in the Table below.

	Te ₂₄ O ₂₉	Te ₃₆ O ₄₃	Te ₄₈ O ₅₈
E _{form} (eV/f.u.)	0.441	0.436	0.438
Te-O Coordination number	2.530	2.555	2.517
E _g (eV) in PBE	0.533	0.485	0.493
E _g (eV) in HSE06	0.989	0.905	0.908

R2Q3. The meaning of the sentence "In fact, V_o is an identical species with Te⁰ and Te²⁺" is unclear. How is an oxygen vacancy identical to neutral or ionized metals in the metal-oxide matrix? Please rewrite this sentence to clarify the meaning.

Our reply: We thank the reviewer's advice. An O atom bridges two Te atoms as in Te-O-Te in TeO₂. A V_o in TeO₂ is formed after removing one O atom in the host Te-O-Te network. Then, Te-Te remains instead

of Te-O-Te. Thus, in terms of understanding, V_o can be considered equivalent to the Te-Te. Te is fully oxidized to Te^{4+} in TeO_2 . With one V_o generation, one Te should be reduced to Te^{2+} for charge neutrality. The number of V_o and Te^{2+} can be the same or $Te^{2+} + Te^{2+} \rightarrow Te^{4+} + Te^0$ can take place. Thus, we mentioned V_o is an identical species with Te^0 and Te^{2+} in the original manuscript.

However, considering the possible misunderstanding of the mentioned sentence and its unimportance to the key core/explanation of this study, we decided to remove this sentence from the revised manuscript.

R2Q4. Was any hysteresis observed in device I-V sweeps? Hysteresis appears to be visible in Fig 3b. Please show forward and reverse sweeps in Fig 3a, Supplementary Figs. 4, 5 and 7, and comment on hysteresis or self-heating effects observed.

Our reply: We thank the reviewer's suggestion. There was hysteresis for the TeO_x -based TFTs and the dual sweep curves were added accordingly (below shows one example in Fig. 3a). The hysteresis stems from charge trapping in the TeO_x -based channel and/or the interfacial traps at the TeO_x/SiO_2 interface. The counterclockwise hysteresis can support this charge-trapping mechanism. Future efforts to improve the semiconducting film and semiconducting/dielectric interface quality are expected to reduce the trap states and the hysteresis.

Figure R5 (Fig. 3a). Transfer characteristics of pristine TeO_x and $TeO_x:Se$ TFTs (both hysteresis direction is counterclockwise).

R2Q5. In the second sentence of the introduction, and again later on page 3 (line 70, line 81), the authors dismiss p-type oxides, with erroneous blanket statements about their VBM composition being made of O 2p orbital. Mention here should be made of various Cu and Sn oxides, which have VBM that are hybridized (Cu 3d and O 2p orbitals; Sn 5s and O 2p), leading to measured hole mobility of $> 100 \text{ cm}^2/\text{Vs}$ in Cu_2O bulk crystals and predicted similarly high hole mobility of stannate oxides, even though thin films have not yet been shown to obtain the same properties. These are interesting materials and active research areas, which compete with the technology demonstrated in this manuscript. See, for example: DOI 10.1021/acs.chemmater.6b03306, 10.1063/1.3231869, <https://doi.org/10.1063/5.0078548>, and 10.1021/acs.chemmater.0c03495.

Our reply: Following the reviewer's good suggestion, we added the corresponding discussion on conventional p-type metal oxides in the revised manuscript with suggested references.

Our revision (page 3): In conventional p-type oxides, *e.g.*, Cu₂O and SnO, the VB orbital hybridization enables good p-type characters.

R2Q6. In Fig. 4, please indicate in the caption the meaning of the red and blue boxes in part (d).

Our reply: We thank the reviewer's comment. The red and blue boxes indicate the position of p-channel TeO_x:Se and n-channel In₂O₃ TFTs, respectively. In the revised manuscript, we added the information in the figure caption.

R2Q7. Please specify in the figure caption or Methods the W/L of the NMOS and PMOS devices used in the inverter, NAND and NOR gates.

Our reply: We thank the reviewer's comment. The W/L of the NMOS and PMOS in inverter and logic gates are 100 μm/60 μm. The corresponding information was added in the revised manuscript (Methods).

R2Q8. For Supplementary Fig 2, please specify the etch time or depth for the XPS traces (*i.e.* are they surface scans or taken after some sputtering has occurred?).

Our reply: We thank the reviewer's comment. The XPS measurement was surface scans without etching to avoid the possible influence on the local bonding environment by sputtering bombardment. The corresponding information was added in the caption of Supplementary Fig. 3. We also provide the XPS depth profile with different etch times in Supplementary Fig. 2.

R2Q9. Tellurium and selenium are well known to be acutely toxic. In order for the work to be repeatable and for the reader to understand possible sources of contamination/impurities, please specify in the Methods sections the safety precautions taken when, for example, loading the materials into the evaporation crucible.

Our reply: We thank the reviewer's comment. In this work, the evaporator was placed inside the N₂-filled glove box. The storage and loading of TeO₂ and Se powder are also performed in the same glove box, which can minimize the toxicity exposure. The corresponding information was added in the revised manuscript (Methods).

Our revision (page 7): The TeO_x:Se films were deposited using a thermal evaporator in a N₂-filled glove box to minimize the possible contaminations and vapor toxicity.

R2Q10. The scale bar in Fig. 1d needs to be enlarged.

Our reply: According to the reviewer's suggestion, we enlarged the scale bar of the corresponding image (now as Fig. 1b), as also shown below.

Figure R6 (Fig. 1b). HRTEM image of 225 °C-annealed TeO_x:Se.

Reviewer #3: The authors bring an interesting study towards amorphous p-type tellurium oxide transistors, which is an important research topic to complement the n-type IGZO transistor for circuit and backplane applications. However, nowadays, complementary circuits are being commercialized, combining IGZO and p-type LTPS transistors, but I would agree to the authors that it would have a clear advantage if a p-type oxide can be found which enables a more simplified process flow. I would recommend to add in the introduction a paragraph on the most promising options to complement the n-type metal-oxide transistors such as p-type LTPS. Despite the fact that it is an interesting topic, from circuit and application perspective, there are several key items missing, of which some of them are listed below.

Our reply to Reviewer 3:

We appreciate and agree with the reviewer's insightful opinion. In the revised manuscript, we added the description of IGZO/LTPS complementary technology and clarified the potential advantages and shortages.

Our revision (page 2): Benefiting from the high mobility and stability, the low-temperature polycrystalline silicon is currently combined with n-type oxides for complementary circuit and backplane applications. Nevertheless, their applications are limited in small-/medium-area devices due to the complex process flow, difficulty in upscaling mass production, and the inhomogeneity from the grain boundary.

Here below we provide a point-by-point response and add the missing items in the revised manuscript.

R3Q1. The characteristics at Fig. 3 show an onset voltage at about +20V V_{GS} , or a positive threshold voltage. This implies that at 0V V_{GS} , there is still substantial current flowing $>10^{-8}$ amps. A complementary circuit will therefore still have considerable leakage current, which is not solving the issue of p-type leakages. A method to control the threshold voltage (onset voltage) at 0V would be very beneficial, to drop the leakage current with a few orders of magnitude, and therefore increase the on/off ratio for a digital circuit.

Our reply: We thank the reviewer's insightful comment. The control of V_{on} to $\sim 0V$ is beneficial to the low-power consumption application. To realize the ideal V_{on} , the channel thickness and charge carrier density need to be fine-tuned. In this study, we explored that the hole source comes from the sub-valence state undercoordinated tellurium, which is closed with the oxygen vacancy generation. To further enable the

adjustment of V_{on} , several approaches are suggested. 1) Exploring alternative film growth techniques, e.g., sputtering would help widely adjust the component stoichiometry and carrier density. 2) Exploring alternative hole suppressors is expected helpful, like the electron suppressors in n-type oxides. 3) Several external processing procedures have been also demonstrated effective to modulate the V_{on} , such as interlayer modification (Phys. Status Solidi RRL, 11, 1700213, 2017), encapsulation (npj 2D Materials and Applications 6, 4, 2022), hydrogen annealing (doi: 10.1002/sdtp.10167), and double gate device configuration (Adv. Mater., 23, 3431, 2011).

We added the information in the revised manuscript (page 6) and a more detailed discussion on V_{TH} modulation below Supplementary Fig. 11.

R3Q2. Applying 40 V on a 100 nm thermally grown SiO_2 is of course feasible, however, the field is very large. This may help to push the mobility if it is dependent on the field. Is this a field-dependent mobility, or is it stable at all fields? Moreover, for a typical application, supply voltages should be kept small, e.g., below 10 V or 5 V. The real on/off ratio of the circuit will therefore degrade. The inverters and logic gates are shown at 20 V; however, the community and readers of this work would be interested how much it scales with downscaling supply voltage.

Our reply: We thank the reviewer’s important comments. The commercial thermally-grown SiO_2 has been one of the most undisputed dielectrics to examine a new semiconductor for TFTs. The field-effect mobility (μ_h) versus V_{GS} is shown below (Figure R7a). The gradually reduced μ_h at high negative V_{GS} region results from the increasingly intense scattering from the interfacial roughness, which is commonly observed in the TFTs based on oxides and other semiconductors (Figure R7b).

Fig. 21. Extracted incremental and average mobilities, μ_{inc} and μ_{avg} , as obtained from experimental I_D - V_{GS} data from a ZnO TFT. Reused with permission from R. L. Hoffman, Journal of Applied Physics, 95, 5813 (2004). Copyright 2004, American Institute of Physics.

Figure R7. (a) μ_h versus V_{GS} curve of one $\text{TeO}_x:\text{Se}$ TFT. (b) One example of ZnO TFT from the Book “Transparent Electronics” (ISBN 978-0-387-72341-9).

The TFT and circuit operation voltage are closely related to the gate capacitance of the dielectric layer. The low-voltage operation can be realized by using a high-permittivity dielectric (Adv. Mater. 30, 1706364, 2018). In the inverters and logic gates, we used 100-nm ALD HfO_2 as the dielectric since the logic gates need to be fabricated on the pre-patterned gate electrodes. In this case, the operation voltage for the TFT and circuit can be lowered to 5 V (as shown below). By reducing HfO_2 thickness, the operation voltage can be reduced more. For example, a 5-nm ALD HfO_2 enabled the TFT operation below 3V (Nat. Electron. 5,

164, 2022). To avoid misunderstanding, we added the dielectric information in the corresponding figure caption. The inverter performance at lower V_{DD} was added in the revised manuscript (Supplementary Figure 11a).

Figure R8 (Supplementary Fig. 11a). Voltage transfer curves of one complementary inverter based on n-channel In_2O_3 and p-channel $\text{TeO}_x\text{:Se}$ TFTs at different V_{DD} .

R3Q3. Another important and missing figure with respect to the oxide TFT circuit are the currents, as indicated in my previous comments. It would be necessary to include in Fig. 4b the supply currents of the inverter. This will be a key figure of merit to qualify the work and proper functioning of the CMOS inverter. The 'real' static leakage current can be calculated from those figures, instead of starting from Fig. 3, which is not indicative.

Our reply: We thank the reviewer for the important comment and advice. We added the inverter current and corresponding statement in the revised manuscript (page 6), also shown below. The static leakage current generally stems from a combined contribution of sub-threshold and gate leakage, both of which are expected to be reduced further by adjusting V_{TH} , reducing TFT I_{off} , reducing power supply, downscaling TFT, etc (Nat. Electron. 1, 30, 2018). More related discussion was also added below in Supplementary Figure 11.

Figure R9 (Supplementary Fig. 11). (a) Voltage transfer curves of one complementary inverter based on n-channel In_2O_3 and p-channel $\text{TeO}_x\text{:Se}$ TFTs at different V_{DD} . (b) Corresponding circuit leakage current.

Our revision (page 6): “The circuit leakage current as a function of V_{DD} was shown in Supplementary Fig. 11. To enable a lower current level, the future efforts on the reduction of power supply, downscaling TFT, and adjustment of the onset voltage of TeO_x:Se TFT at ~0 V would be helpful.”

R3Q4. The circuits have been based on the combination of the studied TeO_x:Se and InO_x transistors. There is only a link to a publication with regards to the InO_x transistors, however, in order to evaluate reproducibility of this work, characteristics of the n-type device would be useful and interesting for the reader. Moreover, the behavior of the InO_x transistors and its on/off ratio are also important for the behavior of the inverter, especially characterized at the dielectric of the inverter/logic gates.

Our reply: We thank the reviewer’s kind comment. More experimental details on the InO_x TFT fabrication were added in the revised manuscript (page 8 Methods). The n-type oxide TFT has been developed maturely in the community with high reproducibility. A typical transfer curve of InO_x TFT on HfO₂ is shown below.

Figure R10. Transfer curve of an n-channel InO_x TFT on the HfO₂ dielectric.

Our revision (page 8, Method): To prepare the InO_x solution, 0.1 M indium nitrate hydrate was dissolved in 2-Methoxyethanol followed by 3 h stirring. The precursor was spun at 5000 rpm for 30 s and then the film was annealed at 250 °C for 0.5 h in air. Finally, Al source and drain electrodes were deposited by thermal evaporation.

R3Q5. The channel lengths of the devices are pretty large and should be indicated in the figures. A 200 μ m channel length is actually not on par with requirements of applications, as those require channel lengths close to 1 μ m. In addition, a 200 μ m channel length results in a large output resistance, which enables a large gain device! Therefore, this figure is a bit misleading. It would be interesting to see the scalability of this parameter at smaller channel lengths.

Our reply: We thank the reviewer’s good suggestion. The channel lengths of 50~300 μ m are generally used geometry in previous research works on oxide TFTs, but we do agree with the reviewer that more downscaling length is interesting and important for the practical integration.

The use of shadow-mask patterning allows the channel length down to 60 μ m and the TFTs exhibit comparable performance (shown below). For the 1 μ m channel length, the photolithography process is

needed, which was demonstrated applicable to our $\text{TeO}_x\text{:Se}$ channel. Narrowing channel length down to 1 μm would usually cause poor gate modulation and contact issues, which are solvable but need more synergistic engineering optimizations of channel thickness, hole concentration, dielectric, and source/drain electrodes, etc, which will be another important work direction in our group in the near future. One additional reason is that the main authors (A.L. and H.Z.) have just moved to Northwestern University for their new positions.

On the other hand, based on the recent success of short-channel n-type oxide TFT with the channel length downscaled to 8 nm (Nat. Electron. 5, 164, 2022), the oxide channel is believed to provide a TFT scaling roadmap for practical integration. We expect this new amorphous p-type semiconductor could open a gate, attracting more joint efforts in the electrical engineering community.

Figure R11. Transfer curve of one p-channel $\text{TeO}_x\text{:Se}$ TFT with different channel lengths of 60 and 200 μm .

R3Q6. Fig. 4 e and f may prove that the NAND/NOR gate is operating correctly, however, the way it is presented can lead to misinterpretation, especially with respect to gain. A better method would be to plot all different characterized curves on top of each other, namely:

- input A = 0 and input B varies from 0V-12V.
- Input B = 0 and input A moves from 0V-12V.
- Input A and B move together from 0V – 12V.

As such, the gain, the switching threshold, the currents and variations with respect to input states will become visible.

Our reply: We thank the reviewer’s good advice. More data with different applied voltages is added to show that the NAND/NOR gate operated correctly (Supplementary Fig. 12, also shown below). We presented them in a widely-used fashion (Nature 562, 254, 2018, Nature Nanotechnology 15, 53, 2020), Nature Electronics 3, 711, 2020). But we do agree that the suggested way would be more specific and would be adopted in the CMOS engineering optimizations in future work.

Figure R12 (Supplementary Fig. 12). The rail-to-rail NAND gate with the working voltage of 10 and 8 V, respectively.

R3Q7. In the section device fabrication, it is written that 100-nm thermally grown SiO₂ was used. However, the TFTs for the logic gates have been made on HfO₂ as gate dielectric. Please make this clear in the text and in the (caption of the) figures. Applying 40 V on this high-k dielectric implies a very large field. The same comment holds for comment (2): how does it scale down, as HfO₂ should enable much lower supply voltage operation, closer to the specifications of the applications. In addition, the characteristics of both TFTs are missing when characterized on HfO₂, which may have a large impact on V_t, on/off ratios and mobilities.

Our reply: We thank the reviewer's suggestion. We added the dielectric information in the corresponding figure caption to avoid misunderstanding (the reason for using different dielectrics is clarified in the above reply, Q3R2). The use of 100-nm HfO₂ in the circuit enables the device operating voltage lower than 20 V due to the larger gate capacitance than SiO₂. In brief, the low-voltage operation for TFT/circuit can be easily realized by using high-k dielectric and/or reducing the dielectric layer thickness.

When we were measuring the logic gates, we observed great rail-to-rail behaviors, which indicated that both n- and p-channel TFTs should perform well; therefore we couldn't feel the need to separately collect the individual TFT data. It was a bit of a pity. But, during the earlier device optimization before the final integration, we did some TFTs on 40-nm HfO₂, which showed good TFT character at a lower operation voltage of less than 10 V (shown below). In future work, more systematic studies on the dielectric effect would be interesting, since different dielectric layers may cause different interface property, channel growth, and the associated electrical parameters, as the reviewer suggested.

Figure R13. Transfer curve of one TeO_x:Se TFT on a 40-nm ALD HfO₂ gate dielectric.

R3Q8. Fig. 3 indicates PBS and NBS with different time durations. Do the authors have an idea about the impact of light on the stability of the devices?

Our reply: We thank the reviewer's question. We measured the light stability under lab fluorescent lamp illumination and the results show a negligible variation in the TFT electrical performance. To eliminate the light illustration-induced TFT instability, the display backplane manufacturing typically adopts the light-shielding layer, which is a mature technology (Information Display, 30, 26, 2014).

Figure R14. Transfer curve of one TeO_x:Se TFT measured under dark and lab fluorescent lamp illumination.

Reviewer #4: The paper describes material properties, density functional calculations and thin film transistor (TFT) characteristics of amorphous evaporated Se-doped TeO_x . The authors demonstrate p-type TFTs with low threshold voltage, high mobility, reasonable stability, manufacturability and complementary circuit integration. There has been a long search for a p-type TFT material that matches the properties of the amorphous n-type metal oxide TFTs, but previous examples have lacked some major aspect of the required properties. The results presented here are therefore interesting. The subthreshold properties are less than ideal and so is the stability but future work may improve these.

Our reply to Reviewer 4:

We sincerely thank the reviewer's insightful comments and evaluation. Here below is our point-by-point response. The manuscript was revised accordingly.

R4Q1. Statistical mobility data is reported for TFTs from one wafer. The paper should also provide information about wafer-to-wafer statistics.

Our reply: We thank the reviewer's suggestion. The statistical device parameters from 4 different batches of TFTs were summarized with good uniformity and reproducibility, as shown below and also added in the revised manuscript.

Figure R15 (Supplementary Fig. 10). c and d exhibit the 80 transfer curves of TeO_x :Se TFTs collected from 4-batch device fabrication and the summarization of corresponding μ_h values.

R4Q2. The statement that V_0 is identical to Te^{2+} (page 4 near bottom) needs some further explanation.

Our reply: We thank the reviewer's advice. An O atom bridges two Te atoms as in Te-O-Te in TeO_2 . A V_o in TeO_2 is formed after removing one O atom in the host Te-O-Te network. Then, Te-Te remains instead of Te-O-Te. Thus, in terms of understanding, V_o can be considered equivalent to the Te-Te. Te is fully oxidized to Te^{4+} in TeO_2 . With one V_o generation, one Te should be reduced to Te^{2+} for charge neutrality. The number of V_o and Te^{2+} can be the same, or $\text{Te}^{2+} + \text{Te}^{2+} \rightarrow \text{Te}^{4+} + \text{Te}^0$ can take place. Thus, we mentioned V_o is an identical species with Te^0 and Te^{2+} in the original manuscript.

However, considering the possible misunderstanding of this sentence and its unimportance to the key core of this study, we decided to remove this sentence from the revised manuscript.

R4Q3. The bias stress instability is attributed to charge trapping rather than defect creation. Does this mean charge trapping in the dielectric, at the interface or in the TeO_x channel, and why?

Our reply: We thank the reviewer's question. For the TFT bias stability test, the robust thermally-growth SiO₂ dielectric was used as dielectric. The charge trapping should happen in the TeO_x-based channel and/or at the TeO_x/SiO₂ interface, which is commonly observed in diverse metal oxide TFTs on SiO₂ dielectric. Moreover, the hysteresis is counterclockwise, which supports the mechanism. In addition, the shifted V_{TH} after bias stress can be recovered quickly within ~20 min, which corresponds to the instability mechanism of charge trapping at or near the dielectric/semiconductor interface (Adv. Mater. 24, 2945, 2012).

R4Q4. The authors claim to have demonstrated manufacturability, but the devices they make are on Si/SiO₂ wafers, rather than using large area technology with a deposited dielectric. While their approach is sufficient for a first publication, any further information they can give about the properties of their TFTs on a deposited oxide would be very valuable.

Our reply: We thank the reviewer's good suggestion. The SiO₂ is the commercial and most accessible/reliable dielectric to examine a new semiconductor for TFTs. In the circuit demonstration, we used 100-nm ALD HfO₂ as the dielectric since the logic gates need to be fabricated on the pre-patterned gate electrodes. The good circuit performance indicates the feasibility of TeO_x on a deposited dielectric. We also provide below a transfer curve of a TeO_x:Se TFT on ALD HfO₂, showing a μ_n of ~10 cm² V⁻¹ s⁻¹ and an I_{on}/I_{off} of ~10⁵. The use of different dielectric layers may influence the channel layer growth and interface property. In future work, specific investigation on this topic would be interesting.

Figure R16. Transfer curve of a TeO_x:Se TFT on a 40-nm ALD HfO₂ gate dielectric.

R4Q5. There is a very recent paper by Choi et al (Electron Device Letters 44, 269, 2023) which describes an alternative TeO TFT made by reactive sputtering. The authors may not have been aware of the paper when they submitted but should include this reference.

Our reply: We thank the reviewer's kind comment. The paper was cited as ref. 29.

The paper is an important addition to TFT technology and the device has a reasonable probability of becoming an industry standard. The paper should be accepted after the authors have responded to the points made above.

Our reply: The authors sincerely thank the reviewer's suggestions and comments, which are very helpful to make this work better.

-----End of the response letter

Reviewer Reports on the First Revision:

Referees' comments:

Please note that Referee #1 was unable to provide a report in this round of revision.

Referee #2 (Remarks to the Author):

The authors have made many improvements to the manuscript which have clarified their methods and results. There are still several points which require amendment:

1. The authors assert in the manuscript that XPS shows "Se atomic ratio (~2%)" and in the response that "We also consulted the XPS manager and confirmed the equipment is capable to measure atomic percentages up to a hundredths decimal point". The fact that XPS software report atomic percentages to 0.01% does not mean they are accurate to this level! XPS is a very precise method, but not accurate.

See, for example, the statement of ~10% accuracy here:

http://www.casaxps.com/help_manual/casaxps2316_manual/xps_spectra.pdf. This is a well known fact and it is surprising that the authors are seemingly unaware of this. Second, atomic ratios are not expressed in percentages. If a percentage is reported, this is an atomic fraction or atomic percentage, not an atomic ratio.

Regarding what COULD be said correctly about Se: the XPS data (Supplementary Fig. 8a) indicates that Se is likely present (the peak in that figure appears to be above the noise level), but it should NOT be used to quantify Se incorporation, as stated above. The most compelling proof of Se incorporation is in the XANES data in Supplementary Fig. 8b-d. Please revise the manuscript accordingly.

2. The manuscript claims the demonstration of "tellurium-oxide-based semiconductors." The narrative of tellurium oxide (of which TeO₂ is the form that readers will automatically assume) does not align with the data, which show that: (a) the Te:O ratio is 1:1.2 according to the authors (but TeO is known to be unstable, so it is unlikely to be a homogenous sub-stoichiometric oxide); (b) XANES shows no detectable Te-O-Te bonds, which should be present if TeO₂ is present; (c) the XANES data show that Se primarily occurs in its metallic state, not its oxidized state. At best the film is a mixed matrix of metallic Te and Se with sub-stoichiometric TeO₂. Claiming this is a TeO₂-like film or TeO_{1.2} film is misleading; it is actually a mixed phase film. The authors need to more clearly state the composition of the film as evidenced by their measurements.

ADDENDUM: additional comments on the response to Reviewer 1

These comments address the authors' response to Reviewer #1's critique.

R1Q1. The EXAFS data (Fig. 1e) clearly show a strong component of Te-Te bonding that is not discussed in the manuscript and does not appear to be part of the ball-stick model of the amorphous film shown in

Supplementary Fig. 1(a). This must be addressed by the authors in the manuscript.

Looking at the incorporation of selenium and analysis in Supplementary Fig. 8, it is clear that the Se is not bonded to the oxygen but is rather present as metallic Se. There is no evidence for formation of Se-O bonds based on this data. The XPS data in Supplementary Fig. 8a is not indicative of the -2 charge state of Se. This peak arises from the Se 3p orbitals. It should have two peaks from spin-orbit coupling and they are weakly dependent on the oxidation state of Se. Inspection of the 3d_{5/2} peaks near 55eV is normally used to look at Se valency. Moreover the XPS data was taken at the surface and thus does not represent the sub-surface bonding of the film.

Based on the materials characterization done here, the film does not consist tellurium oxide but rather of a mixed matrix of sub-stoichiometric TeO_x (x<2), metallic Te, and metallic Se (in the films that incorporate Se). This composition is supported by the optical data (Supplementary Fig 5) which shows that the experimentally realized film has much smaller bandgap (~1.0-1.2eV) compared to the expected TeO₂ bandgap of 3.5eV (this was the topic of R1Q3). Clearly this film is not TeO₂ or even TeO_x. The claim that this is a tellurium oxide film is not well supported by the evidence. The manuscript should be revised to accurately describe the film composition.

R1Q2. Making the model match the film density does not ensure accuracy of the model. The fraction of Te-Te bonds used in the DFT calculation is not clear, and the quoted average coordination number used in DFT is not justified by experimental data.

My comment on R1Q4 can be found below; in brief, the authors' response is inadequate.

R1Q5. The authors' response is adequate.

R1Q6. For publication in Nature, the authors should show results from multiple deposition runs done on different days to document the repeatability of the film synthesis. By using the same deposition process multiple times, the authors should demonstrate repeatability of both the thickness and of the electronic properties (TFT behavior).

Additional comment: The film behavior is very strongly dependent on the "channel post-annealing temperature" as shown in Supplementary Fig. 7. I cannot find any details on this anneal process. When is the post-annealing performed – after Te-O film deposition, after source/drain metal deposition, etc? What ambient and pressure was used for this anneal? Are all the results reported (materials and device characterization) for films that had been annealed in an identical way?

Referee #3 (Remarks to the Author):

The authors have made significant improvements to the manuscript on amorphous p-type tellurium oxide transistors in response to the reviewer's suggestions. However, I still have a couple of suggestions to further enhance the paper.

1. Emphasize the importance of near-zero threshold or onset voltages in future research to help the research community: the added current graphs of the inverters have indicated unexpectedly only a minor current modulation with respect to the input state, presented in supplementary figure 11. This implies that a large static leakage current is present, which diminishes the advantages of having a complementary technology.
2. I would agree to the authors' statement that downscaling the channel length is out of scope for this paper. However, the characterized high gain of the authors cannot be solely attributed to the fact that a complementary technology has been used. One of the main contributions to my opinion is the large channel length. A larger channel length improves the output resistance and consequently impacts positively the gain. It would be more interesting to see the calculation of the noise margin as a function of supply voltage.
3. Can the authors also include the W and L in the appropriate figures, e.g. in Fig. 4a one can write the W/L transistor sizes directly next to the transistor.
4. I apologize for any confusion caused by my previous remark regarding the NAND/NOR gate. My main concern was related to the presentation of the data, rather than the need for additional measurements. To address this, I suggest keeping the plots in Figure S12 and Figures 4e-f, as they provide valuable information. Additionally, it would be helpful if the authors could create a similar plot to Figure 4b to visualize the real transitions. For instance, plotting the NAND output versus input V_{in1} , while keeping V_{in2} at VDD, would generate a plot similar to the inverter but for one of the states of the complex digital gate. This addition will aid in better understanding the behavior of the complex digital gate and enhance the completeness of the results.

Referee #4 (Remarks to the Author):

The authors have given a comprehensive and detailed response to the reviewer comments. In my opinion the responses fully address the points that were made and the changes that they have made to the text respond well to the comments and improve the paper. Regarding the statistics, they have provided appropriate information about wafer-to-wafer variations, as requested. The paper should be accepted for publication.

Author Rebuttals to First Revision:

Reviewer #2: The authors have made many improvements to the manuscript which has clarified their methods and results. There are still several points which require amendment:

Our reply to Reviewer 2:

We genuinely appreciate the reviewer's invaluable comments and suggestions for enhancing our manuscript, particularly the detailed component description. Below is our point-by-point response, and we have revised the manuscript accordingly.

R2Q1. The authors assert in the manuscript that XPS shows "Se atomic ratio (~2%)" and in the response that "We also consulted the XPS manager and confirmed the equipment is capable to measure atomic percentages up to a hundredths decimal point". The fact that XPS software report atomic percentages to 0.01% does not mean they are accurate to this level! XPS is a very precise method, but not accurate. See, for example, the statement of ~10% accuracy here: http://www.casaxps.com/help_manual/casaxps2316_manual/xps_spectra.pdf. This is a well known fact and it is surprising that the authors are seemingly unaware of this. Second, atomic ratios are not expressed in percentages. If a percentage is reported, this is an atomic fraction or atomic percentage, not an atomic ratio.

Our reply: We appreciate the reviewer's insightful comments. We concur with the correction suggested by the reviewer, and the term "Se ratio" has been appropriately revised to "Se percentage". Furthermore, we acknowledge that describing the Se percentage using XPS is inappropriate due to the involvement of unexpected impurities, such as carbon and oxygen elements. These impurities cause a much lower actual Se percentage compared to the host elements, namely Te and/or O.

To achieve a more accurate characterization of the Se incorporation percentage, we conducted high-resolution inductively coupled plasma mass spectrometry (ICP-MS) (Supplementary Fig. 5 and also shown below). The deposited film was dissolved in a mixed acid solvent of HNO₃ and HCl for analysis. The resulting Se atomic percentage was determined to be approximately 25% (Se:Te=1:3). The previous XPS analysis was removed, and the ICP data has been incorporated into the revised manuscript for a more precise characterization of Se percentage.

Supplementary Fig. 5. The high-resolution inductively coupled plasma mass spectrometry spectra of Te and Se characterization elements. The inset picture shows the solution used, which is prepared by dissolving the deposited Se-alloyed TeO_x thin film in mixed HNO_3 and HCl acid.

Regarding what COULD be said correctly about Se: the XPS data (Supplementary Fig. 7a) indicates that Se is likely present (the peak in that figure appears to be above the noise level), but it should NOT be used to quantify Se incorporation, as stated above. The most compelling proof of Se incorporation is in the XANES data in Supplementary Fig. 7b-d. Please revise the manuscript accordingly.

Our reply: We appreciate the reviewer's feedback, and following the suggestion, we have omitted the statement and data regarding Se incorporation using XPS to prevent any potential misunderstanding. Instead, we utilized XANES to validate the Se incorporation through the observation of metallic bonding.

Our revision (page 5): "The EXAFS analysis confirmed the Se incorporation with clear Se-Te bonding (Supplementary Fig. 7)".

R2Q2. The manuscript claims the demonstration of "tellurium-oxide-based semiconductors." The narrative of tellurium oxide (of which TeO_2 is the form that readers will automatically assume) does not align with the data, which show that: (a) the Te:O ratio is 1:1.2 according to the authors (but TeO is known to be unstable, so it is unlikely to be a homogenous sub-stoichiometric oxide);

Our reply: We appreciate the reviewer's correction. Based on the XANES analysis, we have determined that the film consists of a mixed phase, comprising metallic Te and sub-stoichiometric tellurium oxide. In the revised manuscript, we have precisely adjusted the film component description to reflect this as the mixed phase of metallic Te in an amorphous tellurium oxide matrix (i.e., Te-TeO_x) to avoid any potential misinterpretation.

(b) XANES shows no detectable Te-O-Te bonds, which should be present if TeO_2 is present;

Our reply: We thank the reviewer's comment. In crystallized TeO_2 , the combination of shorter Te-O(1) and longer Te-O(2) is well maintained with the long-range ordered arrangement of $-\text{[Te-O(1/2)-Te]}_n-$. The

RDF peak for the combination of Te-O(2) & Te-O-Te is also observed. For the deposited amorphous film, the long-range ordering of $-\text{[Te-O}(1/2)\text{-Te]}_n-$ show rapidly disappears in peaks corresponding to longer Te-O(2) bonds. This means that the crystalline Te-O(2) bonds disappear in the amorphous film, but amorphous Te-O(2) bonds may still exist.

Therefore, it is difficult to clearly state whether the TeO_2 phase is present or not in the deposited film. Instead, the XPS analysis confirmed the presence of TeO_2 (Supplementary Fig. 2). We can understand that the amorphous film contains a large number of short-range disordered TeO_2 rather than the long-chain ones. This short-range disorder is consistent with the previous report on evaporated TeO_2 (J. Phys. Chem. Lett. 2020, 11, 427).

(c) XANES data show that Se primarily occurs in its metallic state, not its oxidized state. At best the film is a mixed matrix of metallic Te and Se with sub-stoichiometric TeO_2 . Claiming this is a TeO_2 -like film or $\text{TeO}_{1.2}$ film is misleading; it is actually a mixed phase film. The authors need to more clearly state the composition of the film as evidenced by their measurements.

Our reply: We agree with the reviewer's opinion. In the revised manuscript, we revised the corresponding statement and clarified that the film composition consists of a mixture phase comprising metallic Te and Se alloy within sub-stoichiometric tellurium oxide to avoid misleading.

ADDENDUM: additional comments on the response to Reviewer 1

RIQ1. The EXAFS data (Fig. 1e) clearly show a strong component of Te-Te bonding that is not discussed in the manuscript and does not appear to be part of the ball-stick model of the amorphous film shown in Supplementary Fig. 1(a). This must be addressed by the authors in the manuscript.

Our reply: We thank the reviewer's good suggestion. In the revised manuscript, we added the discussion on the metallic Te component (Te-Te bonding) in EXAFS section.

Our revision (page 6): "Additionally, noticeable metallic Te-Te bonds were observed, suggesting the spontaneous generation of elemental Te in the final films. This can be related to the redox behaviour of tellurium, in which partial Te^{4+} was reduced to elemental Te in the molten TeO_2 in an inert atmosphere and with tungsten boat reaction".

The amorphous model in Supplementary Fig. 1(a) contains the Te-Te bonding, which was blue-marked in the revised updated figure, as also shown below (middle one). The corresponding description (longer Te-O(2) bond partially broken with the formation of Te-Te metallic bonding) was also added in the figure caption of revised Supplementary Figure 1a for easy assessment. We also include the final Se-alloy model and clarify the Se location and bonding with Te (right one).

Looking at the incorporation of selenium and analysis in Supplementary Fig. 6, it is clear that the Se is not bonded to the oxygen but is rather present as metallic Se. There is no evidence for formation of Se-O bonds based on this data. The XPS data in Supplementary Fig. 6a is not indicative of the -2 charge state of Se. This peak arises from the Se 3p orbitals. It should have two peaks from spin-orbit coupling and they are weakly dependent on the oxidation state of Se. Inspection of the 3d_{5/2} peaks near 55eV is normally used to look at Se valency. Moreover the XPS data was taken at the surface and thus does not represent the sub-surface bonding of the film.

Our reply: We thank the reviewer's professional suggestion. We agree that we cannot clarify the Se valence state from XPS Se 3p orbitals. In the RDF of Se K-edge XAFS (Supplementary Fig. 7b), we can tell that alloyed Se is located at an oxygen-vacant site with the formation of Te-Se-Te bonding rather than Se-O or Se-Se bonding. In the revised manuscript, we removed the Se XPS data and corresponding statement and focused on the discussion on XANES analysis.

Based on the materials characterization done here, the film does not consist tellurium oxide but rather of a mixed matrix of sub-stoichiometric TeO_x ($x < 2$), metallic Te, and metallic Se (in the films that incorporate Se). This composition is supported by the optical data (Supplementary Fig 4) which shows that the experimentally realized film has much smaller bandgap (~1.0-1.2eV) compared to the expected TeO₂ bandgap of 3.5eV (this was the topic of R1Q3). Clearly this film is not TeO₂ or even TeO_x. The claim that this is a tellurium oxide film is not well supported by the evidence. The manuscript should be revised to accurately describe the film composition.

Our reply: We thank the reviewer's comments. Based on the above reply, we know the final film is composed of a mixed phase of sub-stoichiometric metallic Te-Se and TeO_x. The existence of narrow E_g of Te (~0.35 eV) results in the smaller E_g of mixed film than that of the pure TeO₂. In the revised manuscript, we describe the film components with mixed phase clearly and the above explanations were added in Supplementary Fig. 4.

R1Q2. Making the model match the film density does not ensure accuracy of the model. The fraction of Te-Te bonds used in the DFT calculation is not clear, and the quoted average coordination number used in DFT is not justified by experimental data.

Our reply: We thank the reviewer's comment. We measured the film density following the previous 1st reviewer's suggestion and later we found the film density has little influence on the DFT model (band

structure). In our DFT calculation, the Te-Te bonds are involved and the average coordination number of Te with oxygen is calculated to be 2.5, which is close to the measured value of 2.3 from EXAFS by fixing the Debye-Waller factor.

In addition, benefiting from the examination of exact Se alloy ratio during revision, we further performed the DFT calculation of Se-alloyed Te-TeO_x composition and the corresponding data was added in the revised manuscript (Supplementary Fig. 8), also shown below.

Supplementary Fig. 8. **a.** Atomic structure of amorphous Se-alloyed Te-TeO_x generated in DFT (Te:O atomic ratio = 1:1.2; Te:Se atomic ratio = 3:1, film density = 5.6 g/cm³). **b.** Projected DOS of the O 2*p* (red), Te 5*p* (dark yellow), and Se 4*p* (green) states in DFT-PBE.

RIQ6. For publication in Nature, the authors should show results from multiple deposition runs done on different days to document the repeatability of the film synthesis. By using the same deposition process multiple times, the authors should demonstrate repeatability of both the thickness and of the electronic properties (TFT behavior).

Our reply: We thank the reviewer's kind suggestion. As we described in the experimental section, the whole process is easily followed and repeated without any specific treatment/processing. Based on the below comments, we further enriched the Experiments part for more details.

To verify the experimental reproducibility, during the revision process, we asked other colleagues (co-authors of this paper) to repeat the experiment independently and similar results were obtained with good uniformity and reproducibility ($I_{on}/I_{off} > 10^6$ and $\mu_h = 17 \sim 19$ cm²/Vs). One of the measurement figures is as follows.

Additional comment: The film behavior is very strongly dependent on the “channel post-annealing temperature” as shown in Supplementary Fig. 6. I cannot find any details on this anneal process. When is the post-annealing performed – after Te-O film deposition, after source/drain metal deposition, etc? What ambient and pressure was used for this anneal? Are all the results reported (materials and device characterization) for films that had been annealed in an identical way?

Our reply: Following the reviewer’s kind reminder, we have incorporated additional details regarding the post-annealing process in the Experimental section. The channel films underwent post-annealing in ambient air at various temperatures for 30 minutes. Subsequently, the source/drain electrodes were deposited onto the channel layers without undergoing additional annealing. All the results reported (materials and device characterization) for films are processed identically.

Reviewer #3: The authors have made significant improvements to the manuscript on amorphous p-type tellurium oxide transistors in response to the reviewer's suggestions. However, I still have a couple of suggestions to further enhance the paper.

Our reply to Reviewer 3:

We sincerely thank the reviewer’s insightful comments and suggestions for improving our manuscript and pointing out one future direction for the optimization of circuit characterizations. Here below we provide a point-by-point response. The manuscript was revised accordingly.

R3Q1. Emphasize the importance of near-zero threshold or onset voltages in future research to help the research community: the added current graphs of the inverters have indicated unexpectedly only a minor current modulation with respect to the input state, presented in supplementary figure 10. This implies that a large static leakage current is present, which diminishes the advantages of having a complementary technology.

Our reply: We thank the reviewer for the important comment. We agree that in our relatively preliminary CMOS demo, the leakage current is a bit high. The leakage current may come from the depletion operation

mode of p-channel Te-TeO_x TFT. Therefore, we share more directions on the future works on V_{TH} modulation (below Supplementary Fig. 10). The main effort of this study is to propose a new high-mobility amorphous stable p-type semiconductor and demonstrate the potential for TFT application. We expect joint efforts from more professional engineering scientists shortly to fulfill the advantages of CMOS technology and this is also one important research direction in our future works.

R3Q2. I would agree to the authors' statement that downscaling the channel length is out of scope for this paper. However, the characterized high gain of the authors cannot be solely attributed to the fact that a complementary technology has been used. One of the main contributions to my opinion is the large channel length. A larger channel length improves the output resistance and consequently impacts positively the gain. It would be more interesting to see the calculation of the noise margin as a function of supply voltage.

Our reply: We thank the reviewer's understanding and professional opinion. In the current stage, we fairly compared the inverter gain for oxide CMOS inverter with a comparable channel length (Table summarized below, <https://doi.org/10.1080/15980316.2021.1977401>). But we do agree with the reviewer's opinion that a small channel length can provide more objective results and in future work, we will pay more attention to the downscaling short-channel TFT and inverter research. Currently, efforts are underway to implement a bottom-gate/bottom-contact device with a channel length ranging from 5 to 2 micrometers. We anticipate reporting the results of these endeavors soon.

Table 3. Summary of the device structure parameters and device performances for the reported all-oxide-TFT-based CMOS inverters.

Year	n-channel material	p-channel material	W/L (μm) for p and n-channel-TFTs	Gain	V _{dd} (V)	NM _H /NM _L (V)	Power consumption	Ref.
2008	In ₂ O ₃	SnO _x	2000/100, 2000/100	11	10	–	–	[95]
2011	a-IGZO	Cu ₂ O	4000/180, 400/180	120	20	6.0/11.68	–	[96]
	a-IGZO	SnO	20.8/10	4.2	17	9.8/1.0	32 pW	[97]
2012	a-IGZO	SnO	–	1.7	15	–	–	[98]
2013	a-IGZO	SnO	1200/650, 256/256	4.9	17	8.6/1.8	32 pW	[99]
2014	a-IGZO	SnO	500/100, 500/100	3	10	–	–	[100]
	ZnO	SnO	60/30, 30/30	17	10	4.29/4.35	–	[101]
2015	In ₂ O ₃	Cu ₂ O	10/2, 5/10	18	1.5	0.93/0.23	1 nW	[102]
2016	ZnO	SnO	200/30, 40/30	12	12	6.2/3.8	–	[103]
2017	a-IGZO	SnO	2000/60, 2000/60	24	40	20/14.4	12.5 μW	[104]
2018	a-IGZO	SnO	–	112	10	5.22/3.54	–	[105]
	a-IGZO	SnO	80/10, 10/10	142	8	3.55/3.43	4 μW	[106]
	a-IGZO	SnO	60/20, 20/20	226	3	1.79/1.17	241.2 nW	[107]
2019	a-IGZO	NiO _x	200/80, 200/80	10.5	12	5.04/5.16	–	[108]
	a-IGZO	Cu _x O	1000/100, 1000/100	37	60	–	–	[109]
	ZnO	SnO	250/100, 100/100	80	10	3.54/3.54	0.28 μW	[110]
	a-IGZO	SnO	900/10, 150/10	92.4	10	–	–	[111]
2020	a-IGZO	Cu _x O	1000/150, 200/150	14	20	–	–	[112]
	a-IGZO	SnO	300/50, 25/50	50	10	–	–	[113]
	ZTO	Cu ₂ O	7200/80, 3600/80	4.2	20	7/6.5	–	[114]
	a-IGZO	Cu ₂ O	300/50, 400/150	232	70	–	–	[115]
2021	a-IGZO	SnO	30/20, 30/20	370	10	4.8/4.7	–	[116]

⊖a-IGZO: amorphous-In-Ga-Zn-O; ZTO: Zn-Sn-O.

Following the reviewer's suggestion, we plot the noise margin with different supply voltages and the values are ~80%, as shown below.

R3Q3. Can the authors also include the W and L in the appropriate figures, e.g. in Fig. 4a one can write the W/L transistor sizes directly next to the transistor.

Our reply: Following the kind reminding, the W and L have been added accordingly.

R3Q4. I apologize for any confusion caused by my previous remark regarding the NAND/NOR gate. My main concern was related to the presentation of the data, rather than the need for additional measurements. To address this, I suggest keeping the plots in Figure S10 and Figures 4e-f, as they provide valuable information. Additionally, it would be helpful if the authors could create a similar plot to Figure 4b to visualize the real transitions. For instance, plotting the NAND output versus input V_{in1} , while keeping V_{in2} at V_{DD} , would generate a plot similar to the inverter but for one of the states of the complex digital gate. This addition will aid in better understanding the behavior of the complex digital gate and enhance the completeness of the results.

Our reply: We thank the reviewer's good and clear advice. The suggested figure has been added as Supplementary Fig. 12 to better understand the complex logic gate behavior.

Supplementary Fig. 12. Plotting of the NAND output vs input V_{in2} , while V_{in1} was kept at $V_{DD} = 12$ V, for better understanding.

-----End of the response letter

Reviewer Reports on the Second Revision:

Referees' comments:

Referee #3 (Remarks to the Author):

The authors have clearly answered my queries and updated the manuscript accordingly. I do agree to their responses and can therefore accept this work. Indeed, as a minor point, the leakage current of the CMOS inverter is high, but the authors have added a text to underline the importance in future. I am also looking forward to the downscaled research in future, as that will be instrumental for final integration.

Referee #4 (Remarks to the Author):

The authors have again responded constructively to the reviewers' comments, and their description of the atomic structure seems to have improved correspondingly. Not being an expert on atomic structure characterization, I don't have any comments on that aspect of the paper. Details of the TFT performance have not changed materially and based on the reported performance, I continue to recommend that the paper is accepted for publication.

Referee #5 (Remarks to the Author):

The paper presents a study of an amorphous p-type semiconductor composed of selenium-alloyed tellurium in a tellurium sub-oxide matrix and its use in p-channel TFTs and complementary circuits. The material was characterised by XRD, HRTEM, XPS and X-ray absorption spectroscopy (XANES/EXAFS). MD-DFT simulations were used to generate the structural model of amorphous Te-TeO_x. The p-channel TFT devices were fabricated and characterised.

The paper suffers from a superficial analysis of X-ray absorption spectra which does not support the conclusions discussed in the text. The rigorous analysis of EXAFS data is fully absent. Therefore, the paper cannot be recommended for publication in the Nature journal in its present form and requires major revision.

The weak points are listed below:

- 1) The Fourier transform (FT) of the EXAFS spectrum (page 2, Fig. 1, Supplementary Fig. 7) is not a radial distribution function (RDF) because it contains contributions from atomic scattering amplitude and phase functions as well as many-atom distributions known as multiple-scattering effects. Therefore, the text and figure captions (also in the Supplementary material) should be corrected.

2) Since the FT of the EXAFS spectrum is not an RDF, it cannot be used to determine any structural information. Instead, simulations of EXAFS spectra must be performed using the available software. Since the authors intend to analyse also the contributions from outer coordination shells up to 4-5 Å (as Te-O-Te), the multiple-scattering effects related to long scattering paths as well as static and thermal disorder effects should be properly considered in the EXAFS simulations.

3) The original EXAFS spectra as well as the imaginary part of FTs are not reported, which makes their comparison ambiguous and does not allow the reader to evaluate data quality. These as well as examples of the EXAFS spectra simulations should be added to the manuscript or supplementary material.

4) The origin of features in XANES spectra in Fig. 1d and Supplementary 7a is not discussed, and their theoretical interpretation is not provided.

5) The origin of the conclusion "The average coordination number of Te with oxygen is calculated to be 2.5, closely aligning with the measured value of 2.3 from EXAFS." on page 3 is unclear, since no analysis of the EXAFS spectra is presented.

6) The conclusion "EXAFS analysis confirmed the existence of Se in Se-Te metallic bonds, indicating that the Se was alloyed into the Te in Te-TeO_x (Supplementary Fig. 7)." is not supported by any simulations/analysis.

7) The experimental details (sample preparation, detectors and mode used, monochromator used, etc), as well as the methodology of XANES/EXAFS analysis (software, approximations used, etc) are not given and should be described in detail in the Methods section.

8) The structure generated by MD simulations can be used to calculate the theoretical EXAFS spectra, which can be further compared to the experimental EXAFS data to validate the quality of MD simulations.

Author Rebuttals to Second Revision:

Reviewer #5: The paper presents a study of an amorphous p-type semiconductor composed of selenium-alloyed tellurium in a tellurium sub-oxide matrix and its use in p-channel TFTs and complementary circuits. The material was characterised by XRD, HRTEM, XPS and X-ray absorption spectroscopy (XANES/EXAFS). MD-DFT simulations were used to generate the structural model of amorphous Te-TeO_x. The p-channel TFT devices were fabricated and characterised.

The paper suffers from a superficial analysis of X-ray absorption spectra which does not support the conclusions discussed in the text. The rigorous analysis of EXAFS data is fully absent. Therefore, the paper cannot be recommended for publication in the Nature journal in its present form and requires major revision. The weak points are listed below:

Our reply to Reviewer 5:

We are grateful for the reviewer's comments and clear suggestions. In earlier revisions, we carried out a thorough series of experiments and collected extensive XANES/EXAFS data. However, we only included some key data requested by the reviewers to aid in understanding the local atomic structure of the material (similar to other film characterisations). We regret this oversight/misunderstanding and, in accordance with the new reviewer's clear guidance, we have incorporated a comprehensive analysis of the data/analysis in the revised manuscript and Supplementary Methods. Below, we respond to the comments point-by-point:

R5Q1. The Fourier transform (FT) of the EXAFS spectrum (page 2, Fig. 1, Supplementary Fig. 7) is not a radial distribution function (RDF) because it contains contributions from atomic scattering amplitude and phase functions as well as many-atom distributions known as multiple-scattering effects. Therefore, the text and figure captions (also in the Supplementary material) should be corrected.

Our reply: We appreciate the reviewer's kind reminder. The expression for the radial distribution function (RDF) has been corrected to Fourier transform (FT) in the main text and figure captions.

R5Q2. Since the FT of the EXAFS spectrum is not an RDF, it cannot be used to determine any structural information. Instead, simulations of EXAFS spectra must be performed using the available software. Since the authors intend to analyse also the contributions from outer coordination shells up to 4-5 Å (as Te-O-Te), the multiple-scattering effects related to long scattering paths as well as static and thermal disorder effects should be properly considered in the EXAFS simulations.

Our reply: We appreciate the reviewer's suggestions. We carried out quantitative analysis using the standard EXAFS procedure, and simulations of EXAFS spectra have been performed for the structural analysis. The results show that using three kinds of chemical bonds around the central Te element for the Se-alloyed Te-TeO_x, Te-O (with oxygen vacancy), Te-Se, and Te-Te, the structural model is well-fitted after EXAFS simulations (Figures R1-1 and R1-2). More details are provided below, including 1) Model structures; 2) Experimental and simulated EXAFS spectra; 3) Parameter tables.

Regarding the second comment, as shown in the FT of the Te K-edge EXAFS, there are no effective FT peaks higher than ~3 Å for long-range ordering of Te-O-Te(Se) in our deposited Se-alloyed Te-TeO_x, indicating an amorphous and locally disordered structure. Therefore, during XAFS analysis, we

approximately considered only the first shells around central Te and Se atoms. Limited to first-shell consideration, only single-scattering paths for central Te and Se atoms could be applied in the structural modeling.

1) Model structure and reference materials: metallic Te, oxide TeO₂, metallic Se, and metallic-like TeSe₂.

For possible atomic scatterings in the structural modeling of Se-alloyed Te-TeO_x, the chemical bonds of Te-O (followed by oxygen vacancy), metallic-like Te-Se, and Te-Te have been preliminarily considered for EXAFS simulation. The theoretical single scattering paths have been calculated with reference materials including TeO₂, TeSe₂, and metallic Te & Se, respectively. Specifically, for the broad second FT peak around 2.5 Å of the Te K-edge FT and the broad first FT peak around 2.3 Å of the Se K-edge FT, two kinds of chemical bonds, Te-Se/Te-Te and Se-Se/Se-Te, have been introduced, respectively.

2) **Experimental and simulated EXAFS spectra:** Based on the proposed structural model above, Figures R1-1 and R1-2 show the experimental and best-fitted FT features, and corresponding inverse Fourier-transformed EXAFS spectra after the curve-fitting process at Te K-edge and Se K-edge, respectively. Also, Tables S1 and S2 present the EXAFS structural parameters for Te K-edge and Se K-edge k^3 -weighted EXAFS spectra of Se-alloyed Te-TeO_x sample, respectively. In order to obtain the total amplitude reduction factor, S_0^2 , the first shell coordination numbers of metallic Te and TeO₂ are fixed to 2. And each value in parentheses means the uncertainty evaluated from the calculation process.

Figure R1-1. (Left) Experimental (black solid line) and best-fitted FT features (blue dash line) for the Se-alloyed Te-TeO_x, and (Right) corresponding inverse Fourier-transformed Te K-edge EXAFS spectra after curve-fitting process, and reference materials of Te and TeO₂.

Figure R1-2. (Left) Experimental (black solid line) and best-fitted FT features (blue dash line) for the Se-alloyed Te-TeO_x, and (Right) corresponding inverse Fourier-transformed Se K-edge EXAFS spectra after curve-fitting process, and reference material of metallic Se powder.

3) Parameters information:

Table S1. EXAFS Structural parameters for Te K-edge k^3 -weighted EXAFS spectra of Se-alloyed Te-TeO_x sample calculated from EXAFS curve-fitting process

Sample	Path	Energy shift (eV)	Coordination number	Interatomic distance (Å)	Debye-Waller factor (10^{-3}Å^2)	r-factor of fit**
Te metallic	Te-Te	5.54 (± 0.74)	2.00*	2.831 (± 0.003)**	5.03 (± 0.11)	0.0023
TeO ₂	Te-O1	6.45 (± 0.85)	2.00*	1.861 (± 0.004)	2.22 (± 0.26)	0.0057
	Te-O2		2.00*	2.108 (± 0.005)	4.15 (± 0.47)	
TeO _x :Se Film	Te-O1	4.73 (± 1.55)	1.44 (± 0.06)	1.878 (± 0.006)	3.64 (± 0.27)	0.0065
	Te-Se		0.34 (± 0.04)	2.576 (± 0.010)	2.74 (± 0.59)	
	Te-Te		0.84 (± 0.06)	2.814 (± 0.008)	4.35 (± 0.36)	

*In order to obtain the total amplitude reduction factor, S_0^2 , the first shell coordination number of metallic Te is fixed to 2. And each value in parentheses means the uncertainty obtained from the calculation process.

** R-factor value which is quality of the fit determined with $\{Re\Delta\chi_k^2 + Im\Delta\chi_k^2\} / \{Re(\chi_{kdata})^2 + Im(\chi_{kdata})^2\}$, where $\chi(k)$ is EXAFS-function) and $\Delta\chi(k)$ means $\chi(k)_{data} - \chi(k)_{best-fitted}$.

Table S2. EXAFS Structural parameters for Se K-edge k^3 -weighted EXAFS spectra of Se-alloyed Te-TeO_x sample calculated from EXAFS curve-fitting process

Sample	Path	Energy shift (eV)	Coordination number	Interatomic distance (Å)	Debye-Waller factor (10^{-3}Å^2)	r-factor of fit***
Se metallic	Se-Se	5.59 (± 0.67)	2.00*	2.381 (± 0.003)**	4.52 (± 0.10)	0.0027
TeO _x :Se Film	Se-Se	7.82 (± 1.31)	0.44 (± 0.06)	2.394 (± 0.010)	2.03 (± 0.52)	0.0051
	Se-Te		0.85 (± 0.08)	2.605 (± 0.008)	2.55 (± 0.29)	

*In order to obtain the total amplitude reduction factor, S_0^2 , the first shell coordination number of metallic Se is fixed to 2. And each value in parentheses means the uncertainty obtained from the calculation process.
** R -factor value which is quality of the fit determined with $\{Re\Delta\chi_k^2 + Im\Delta\chi_k^2\} / \{Re(\chi_{kdata})^2 + Im(\chi_{kdata})^2\}$, where $\chi(k)$ is EXAFS-function) and $\Delta\chi(k)$ means $\chi(k)_{data} - \chi(k)_{best-fitted}$.

The above analysis and data were added in the revised manuscript (Supplementary Methods in SI, page 8-11, Figure S12 and Table 2-3)

R5Q3. The original EXAFS spectra as well as the imaginary part of FTs are not reported, which makes their comparison ambiguous and does not allow the reader to evaluate data quality. These as well as examples of the EXAFS spectra simulations should be added to the manuscript or supplementary material.

Our reply: We appreciate the reviewer's suggestions. In addition to the original EXAFS data, the real and imaginary parts of FTs and EXAFS spectra are provided in **Figure R2**. EXAFS simulation results are presented in the previous response R5Q2.

The Te and Se K-edge k^3 -weighted EXAFS spectra for Se-alloyed Te-TeO_x exhibit well-defined spectral features even in high k -space. In the EXAFS analysis, the k -space regions between 3.5 and 14 Å (for Te K-edge) and 15 Å (for Se K-edge) have been used.

Figure R2. (Upper) Experimental k^3 -weighted EXAFS spectra and (Bottom) Fourier-transformed magnitudes including real and imaginary parts at the (left) Te K-edge and (right) Se K-edge for the Se-alloyed Te-TeO_x, compared to those of reference Te and Se materials.

The above analysis and data were added in the revised manuscript (Supplementary Methods in SI, page 11, Figure S13).

R5Q4. The origin of features in XANES spectra in Fig. 1d and Supplementary 7a is not discussed, and their theoretical interpretation is not provided.

Our reply: Thanks for the reminder. The XANES spectra in Fig. 1d and Supplementary 7a are measured results. For the reference Te and TeO₂, white line spectral features at ~31820 eV arise from an electric-dipole-allowed transition of the 1s electron to the unoccupied 5p orbital with the electronic configurations of [Kr]4d¹⁰5s²5p⁴ state and [Kr]4d¹⁰5s²5p⁰ state (Te(IV)), respectively. The Te K-edge XANES feature for the Se-alloyed Te-TeO_x exhibits mixed characteristics resembling both metallic Te and oxide TeO₂ reference materials. Here, we tried to ascertain the chemical composition ratio of these two phases in the Se-alloyed Te-TeO_x using a linear combination of the reference spectra. As a result, we determined an averaged composition ratio of approximately 4:6 for metallic Te and TeO₂ phases.

Figure R3. Spectral comparison of Se-alloyed Te-TeO_x after linear combination with metallic Te and oxide TeO₂.

The above discussions were added in the revised manuscript (Supplementary Methods in SI, page 11)

R5Q5. The origin of the conclusion "The average coordination number of Te with oxygen is calculated to be 2.5, closely aligning with the measured value of 2.3 from EXAFS." on page 3 is unclear, since no analysis of the EXAFS spectra is presented.

Our reply: We appreciate the reviewer's reminder. The previous statements are imprecise and were corrected in the revised manuscript.

As listed in Table R1, the total coordination number around the central Te atom for the Se-alloyed Te-TeO_x from EXAFS is about 2.6, as the summation of 1.44 (± 0.02) for Te-O, 0.35 (± 0.01) for Te-Se, and 0.84 (± 0.01) for Te-Te. For the DFT calculation, the average coordination number of Te with oxygen is 2.5. In detail, Figure R4 shows the calculated O radial distribution function (RDF) and accumulated O RDF (A-RDF) around Te for the representative sample of the Se-alloyed Te-TeO_x amorphous structures obtained in DFT MD simulations. The first Te-O shell is clearly shown at around 2 Å. The Te-O coordination number can be obtained as the A-RDF value at the first local minimum in RDF beyond the first Te-O shell, and the ensemble-averaged value is 2.5. This general approach includes both the Te-O1 bonds and the inter-TeO₂-molecular Te-O2 bonds, and can be larger than the EXAFS value of 1.44 (Table R1), which includes only the Te-O1 intra-molecular bonds. Therefore, direct comparison of the Te-O coordination numbers between calculation and EXAFS is limited. Consequently, we have removed the expression "closely aligning with the measured value of 2.3 from EXAFS" from the main text.

Figure R4. Calculated O RDF (black) and accumulated O RDF (red) around Te in Se-alloyed Te-TeO_x amorphous structures obtained in DFT MD simulations.

R5Q6. The conclusion "EXAFS analysis confirmed the existence of Se in Se-Te metallic bonds, indicating that the Se was alloyed into the Te in Te-TeO_x (Supplementary Fig. 7)." is not supported by any simulations/analysis.

Our reply: We appreciate the reviewer's reminder. As depicted in Figure R5, the broad FT peaks corresponding to Se-Te/Te-Se in TeO₂:Se are situated between the FT peak of pure Se-Se and that of pure

Te-Te. This intermediate FT peak feature suggests the presence of Se-Te (or Te-Se) chemical interaction in the amorphous Se-alloyed Te-TeO_x. Meanwhile, as a reply in Q2, the chemical bonding of Se-Te (or Te-Se) in Se-alloyed Te-TeO_x was incorporated during EXAFS calculation, resulting in a better r-factor of fit. The high alloy capability of Se and Te was also demonstrated in the previous studies.^[1,2]

Figure R5. Comparison of Fourier-transformed features for Te K-edge (black solid line) and Se K-edge (red solid line) EXAFS spectra, compared to those of reference metallic Te and Se powder (dashed line).

Figure R5 was added as new Supplementary Fig. 7c and the corresponding discussions were added below the Supplementary Fig. 7.

R5Q7. The experimental details (sample preparation, detectors and mode used, monochromator used, etc), as well as the methodology of XANES/EXAFS analysis (software, approximations used, etc) are not given and should be described in detail in the Methods section.

Our reply: According to the reviewer’s suggestion, we added more details in Supplementary Methods for XANES/EXAFS analysis (page 11-13).

X-ray absorption spectroscopy: Te and Se K-edge X-ray absorption spectra of the Se-alloyed Te-TeO_x film, X-ray absorption near edge structure (XANES), and extended X-ray absorption fine structure (EXAFS), were collected on BL10C beam line (using multiple wiggler source) at the Pohang light source (PLS-II) with top-up mode operation under a ring current of 250 mA at 3.0 GeV. The monochromatic X-ray beam could be obtained using liquid-nitrogen cooled Si(311) double crystal monochromator (Bruker ASC). For Te (31814 eV) and Se (12658 eV) K-edge XAFS measurements, X-ray absorption spectroscopic data were recorded in fluorescence mode with 7 7-channel silicon drift detector (SDD, Rayspec Ltd.) as a photon detector. Higher-order harmonic contaminations were eliminated by detuning to reduce the incident X-ray intensity by ~20%. Energy calibration has been carried out with reference Te and Se metal powders.

XAFS data analysis: The XAFS data analysis was performed through the standard XAFS procedure.¹⁻⁴⁾ Using AUTOBK module in UWXAFS package⁵⁾, the k^3 -weighted Te K-edge and Se K-edge EXAFS spectra, $k^3\chi(k)$, have been obtained through background removal and normalization processes. The $k^3\chi(k)$ spectra have been Fourier-transformed (FT) in the k ranges between 3.5 and 14.0 Å⁻¹ (Te K-edge) and 15.0 Å⁻¹ (Se K-edge). The experimental FT spectra have been inversely Fourier-transformed with the *hanning* window function in the r space range between 1.0 and 3.2 Å (Te K-edge) and 3.0 Å⁻¹ (Se K-edge). To

determine the EXAFS structural parameters for the first bond pairs, the curve-fitting process has been carried out by using the single bonding model. Theoretical single scattering paths of the first shells around the central Te element have been calculated with FEFF9 code⁽⁶⁻⁷⁾ under the space groups of $P 41 21 2$ for the tetragonal TeO₂ model, $P 31 2 1$ for the trigonal Te metallic, $C2$ for the monoclinic TeSe₂, $P 31 2 1$ for the trigonal Se metallic. In the EXAFS curve fitting process with the FEFFIT module, the total amplitude reduction factor, S_0^2 , was fixed to 0.7 for the Te K-edge XAFS and 0.9 for the Se K-edge XAFS, which were obtained after EXAFS fitting for metallic Te and Se EXAFS spectra with constant two coordination numbers. The EXAFS structural parameters, interatomic distance (r), coordination numbers (N), Debye-Waller factor (σ^2), have been determined within the allowed R -factor value which is the quality of the fit with $\{Re\Delta\chi_k^2 + Im\Delta\chi_k^2\} / \{Re(\chi_{kdata})^2 + Im(\chi_{kdata})^2\}$, where $\chi(k)$ is EXAFS-function) and $\Delta\chi(k)$ means $\chi(k)_{data} - \chi(k)_{best-fitted}$. For k - r space correlations, *Morlet* wavelet-transformed EXAFS have been also obtained with proper values of η and σ in equation spectra⁽⁸⁻⁹⁾ as follows;

$$\psi(t) = \frac{1}{\sqrt{2\pi}\sigma} (e^{i\eta t} - e^{-\eta^2 \sigma^2 / 2}) e^{-t^2 / 2\sigma^2}$$

where the η is the frequency of the oscillation functions and the σ is the half width.

1. J. J. Rehr and R. C. Albers, Theoretical Approaches to X-ray Absorption Fine Structure, *Rev. Mod. Phys.* 72, 621, (2000).
2. J.J. Rehr and R.C. Albers, Multiple Scattering theory: Scattering-matrix formulation of curved-wave multiple-scattering theory: Application to x-ray-absorption fine structure, *Phys. Rev. B* 41, 8139 (1990).
3. M. Newville, IFEFFIT: interactive EXAFS analysis and FEFF fitting, *J. Synchrotron Rad.* 8, 322 (2001).
4. ATHENA, ARTEMIS, HEPHAESTUS: data analysis for X-ray absorption spectroscopy using IFEFFIT, B. Ravel and M. Newville, *J. Synchrotron Rad.* 12, 537 (2005).
5. E.A. Stern *, M. Newville, B. Ravel, Y. Yacoby, D. Haskel, The UWXAFS analysis package: philosophy and details, *Physica B: Condensed Matter*, 208-209, 117, 1995.
6. J. J. Rehr, J. J. Kas, F. D. Vila, M. P. Prange, K. Jorissen, Parameter-free calculations of X-ray spectra with FEFF9. *Phys. Chem. Chem. Phys.*, 12, 5503- 5513, 2010.
7. A.L. Ankudinov, B. Ravel, J.J. Rehr, and S.D. Conradson, FEFF8: Real Space Multiple Scattering Calculation of XANES, *Phys. Rev. B* 58, 7565, 1998.
8. H. Funke*, A. C. Scheinost, Wavelet analysis of extended x-ray absorption fine structure data, *PHYSICAL REVIEW B*, 71, 094110, 2005.
9. H. Funke*, M. Chukalina, A. C. Scheinost, A new FEFF-based wavelet for EXAFS data analysis, *J. Synchrotron Rad.* 14, 426-432, 2007.

R5Q8. The structure generated by MD simulations can be used to calculate the theoretical EXAFS spectra, which can be further compared to the experimental EXAFS data to validate the quality of MD simulations.

Our reply: We appreciate the reviewer's suggestion.

As shown in the figure below, the structure generated by MD simulations is highly complex, with numerous Te sites exhibiting different local site symmetries. Consequently, performing theoretical EXAFS calculations for the DFT model becomes a very challenging task.

Specifically, the structural model suggested in the EXAFS simulation comprises detailed features: 1) partially-broken longer Te-O bonds followed by oxygen vacancy sites, 2) formation of Te-Te metallic bonding (under local distortion around the central Te element), 3) incorporation of doped Se into oxygen vacancy sites, along with locally-formed Te-Se bonds; Meanwhile, the DFT model illustrates a broad atomic distribution, including Te-O (with oxygen vacancies), Te-Se, Te-Te, and Se-Se chemical bonds. In summary, these important findings for the Se-alloyed Te-TeO_x could support the overall distribution of Te-O (with oxygen vacancy), Te-Se, Te-Te, and Se-Se bonding pairs presented in the DFT-MD study.

Authors' statements

We would like to sincerely thank the reviewer again for the clear suggestions, which are very helpful for us to improve the XANES/EXAFS analysis and discussion. In this study, including XANES/EXAFS, a comprehensive series of film characterizations have been carefully conducted, the information regarding the components and the local atomic structure has been obtained, and eventually, the macroscopically amorphous structure of this new compound has been elucidated; The DFT analysis also helped understand the electrical band structure. Meanwhile, the key achievement of this work lies in the successful development of the long-awaited first amorphous high-performance, stable, and commercially compatible p-channel transistors. This accomplishment represents a milestone for large-area CMOS technology, particularly considering the prolonged anticipation since the invention/commercialisation of the amorphous n-channel TFT in the mid-2000s (*Nature* 432, 488-492, 2004).

References

- [1] Tan, C. *et al.* Evaporated $\text{Sb}_{1-x}\text{Te}_x$ thin films with tunable bandgaps for short-wave infrared photodetectors. *Adv. Mater.* **32**, 2001329 (2020).
- [2] Hadar, I., Hu, X., Luo, Z.-Z., Dravid, V. P. & Kanatzidis, M. G. Nonlinear band gap tunability in selenium–tellurium alloys and its utilization in solar cells. *ACS Energy Lett.* **4**, 2137-2143 (2019).

-----End of the response letter

Reviewer Reports on the Third Revision:

Referees' comments:

Referee #5 (Remarks to the Author):

The revision is satisfactory. I recommend accepting the manuscript for publication.